# Fisher-Preserving Guidance: Training-Free Manifold Constraints for Safe Diffusion Control

**Hao Ren** [1] **Zetong Bi** [1] **Yiming Zeng** [1] **Le Zheng** [1] **Zhi Li** [1] **Zhaoliang Wan** [2] **Lu Qi** [2] **Hui Cheng** [1]

## Abstract

Diffusion models are effective for waypoint prediction in visual navigation, but standard sampling and test time guidance can produce unreliable or inefficient trajectories when updates drift off the training manifold. We propose Fisher Preserving Guidance with Outer Product Span Projection, a training-free inference method that avoids large Fisher drift associated with off-distribution actions while optimizing a task objective. Our method computes the Fisher-preserving update via a low-rank Jacobian factorization, requiring only a single backward pass per step and enabling real-time use. We further introduce Truncated Fisher Denoising Sensitivity as an uncertainty signal and use it for robust multi-sample action blending. Experiments on toy and realistic navigation benchmarks, including Maze2D with TSDF-based guidance, PushT with official Diffusion Policy weights, and visual navigation in simulation and on real robots, demonstrate consistent improvements in performance over strong diffusion-policy baselines without additional training.

## 1. Introduction

Visual navigation is a fundamental capability for embodied agents such as mobile robots and autonomous vehicles. The primary challenge is converting first-person image streams into accurate, robust waypoint predictions and temporal distance estimates while accounting for perception noise, dynamic obstacles, and the multi-modal nature of real-world environments. Traditional methods rely on spatial information such as point clouds, making them difficult to transfer to RGB-only visual navigation tasks (Oriolo et al., 1995; Chalvatzaras et al., 2022). In recent years, end-to-end deep learning approaches for visual navigation trajectory planning have made notable progress (Zhu et al., 2017; Beeching et al., 2020; Chen et al., 2021). However, regression-based models struggle to capture multi-modal action distributions, leading to sub-optimal or erroneous actions (Chi et al., 2023; Xing et al., 2025; Li et al., 2025).

Recent advances in generative modeling, particularly diffusion models, have shown great promise for this task (Sridhar et al., 2024; Gode et al., 2024; Ren et al., 2025). Diffusion-based policies can generate diverse and expressive waypoint distributions, naturally capturing the uncertainty inherent in navigation and supporting flexible, sample-based decision making. However, ensuring that these sampled actions are both reliable and safe in complex, ambiguous scenes remains a significant challenge. In practice, sampling methods guided solely by task loss often yield predicted waypoints far from the training data, leading to sub-optimal or out-of-distribution behaviors (Filos et al., 2020; Yang et al., 2024). Meanwhile, the rich uncertainty structure encoded by diffusion processes is rarely fully leveraged. The diffusion sampler generates multiple plausible actions, yet most systems either pick one randomly or rely on hand-tuned heuristics (Janner et al., 2022; Chi et al., 2023; Sridhar et al., 2024).

Ideally, a navigation policy should constrain its sampling trajectory to remain in domain, well-understood regions of the action space, those supported by the training data or extra constraints during inference, while still allowing flexible guidance toward the goal. Achieving this without sacrificing efficiency has proven challenging (Sun & Song, 2025).

To address these limitations, we propose Fisher-Preserving Guidance with Outer Product Span projection (FPG-OPS), an efficient inference framework for diffusion-based visual navigation. FPG-OPS constrains each reverse-diffusion step to lie on a Fisher isosurface by using Outer Product Span projection, thus ensuring the sampling trajectory remains near the data manifold while simultaneously optimizing task objectives like path efficiency. Leveraging the low-rank structure of the model's residual head, our method computes the required Fisher-preserving update in a low-dimensional latent space with a single backward pass, reducing complex-

---

[1]Sun Yat-sen University, Guangzhou, China [2]Insta360 Research, Shenzhen, China. Correspondence to: Hui Cheng <chengh9@mail.sysu.edu.cn>.

*Proceedings of the 43rd International Conference on Machine Learning*, Seoul, South Korea. PMLR 306, 2026. Copyright 2026 by the author(s).

ity by two orders of magnitude compared to full-rank Fisher computation. Additionally, we propose an action blending strategy based on Truncated Fisher Denoising Sensitivity (TFDS) and cluster typicality, jointly leveraging model uncertainty and distributional consensus for robust waypoint selection. This framework can be seamlessly integrated into existing diffusion-based methods without retraining, accelerating inference while maintaining model performance, and enabling principled selection and blending of multi-modal actions based on TFDS. Our contributions are as follows:

**1)** We propose a training-free Fisher-preserving guidance method that enforces first-order Fisher consistency at each step, preventing off-manifold drift and enhancing reliability.

**2)** We develop a low-rank Outer-Product-Span projection method that enables efficient, real-time computation of the Fisher-preserving update with minimal overhead.

**3)** We introduce an uncertainty-guided action blending mechanism based on truncated Fisher sensitivity and cluster typicality, improving the robustness and efficiency of diffusion policy navigation.

## 2. Related Work

**Visual Navigation.** Visual navigation is an important and enduring challenge in robotics and embodied AI, demanding that agents transform raw sensory observations into sequential actions for goal-directed navigation. Classical approaches typically decompose this problem into modular components: visual mapping, metric or topological localization, and explicit path planning (Cadena et al., 2016; Yang et al., 2016; Yasuda et al., 2020; Zheng et al., 2025). While effective in structured settings, these pipelines rely heavily on accurate perception and are often brittle in the face of sensor noise, perceptual aliasing, or accumulated localization errors over long horizons (Hu et al., 2023).

With the rise of deep learning, end-to-end visuomotor policies have become popular, bypassing explicit mapping and directly predicting actions from image inputs (Zhu et al., 2017; Chen et al., 2021; Majumdar et al., 2022; Al-Halah et al., 2022; Wu et al., 2020; Wan et al., 2026). There has been extensive research in related fields on vision-and-language navigation (Zhou et al., 2024; Li et al., 2024) and target-driven navigation (Xie et al., 2025; Tang et al., 2022). This paper focuses specifically on RGB-only visual navigation. Recent work has introduced powerful architectures such as topological memory networks ViNT (Shah et al., 2023b), conditional diffusion-based planners NoMaD (Sridhar et al., 2024), NaviDiffusor (Zeng et al., 2025), Flow-Nav (Gode et al., 2024) and prior injection diffusion policy NaviBridger (Ren et al., 2025). These advances have improved performance and generalization, particularly in unfamiliar or visually complex environments. However,

despite these successes, existing methods often overlook a crucial aspect of robust navigation, the ability to reason about the credibility and diversity of candidate actions generated by stochastic policies (Du et al., 2021; Shah et al., 2023a; Sridhar et al., 2024). In most current systems, action selection is based on random sampling or simple confidence metrics, without explicitly modeling the typicality of an action within the policy's generative distribution or its stability under observation perturbations. This can lead to weak decisions, poor action fusion, and limited robustness to out-of-distribution inputs.

**Fisher Information and Guided Diffusion.** Recent work has highlighted the centrality of uncertainty quantification in robust deep learning and generative modeling (Kendall & Gal, 2017; Gal & Ghahramani, 2016; Song & Lai, 2024). In diffusion models, Fisher information and Jacobian-based sensitivity have emerged as useful tools for understanding sample reliability and calibration (Zheng et al., 2023; Deng et al., 2023). Several approaches regularize or control Fisher information during training or inference to encourage smoother mappings and more stable predictions (Song & Lai, 2024; Gao et al., 2024). Directly computing or enforcing Fisher- or Jacobian-based constraints, however, is often prohibitively expensive in high-dimensional vision and control settings (Jiang et al., 2024; Kou et al., 2023; Deng et al., 2023). Prior work improves scalability through low-rank approximations and efficient estimators, including Hutchinson-style trace estimation (Hutchinson, 1989) and latent-space factorizations (Wang et al., 2025; Song & Lai, 2024).

Our work is also related to training-free guided diffusion and posterior sampling, where the reverse diffusion process is modified by external guidance. A representative example is Manifold Preserving Guided Diffusion (MPGD) (He et al., 2024), which reduces off-manifold drift in guided posterior sampling through manifold-consistent guidance and latent/on-manifold projections. While sharing the general goal of structure-preserving guidance, our work focuses on diffusion control rather than conditional generation or posterior sampling. We introduce a Fisher-sensitivity-preserving projection and TFDS-based action blending to stabilize action generation and selection in RGB visual navigation, providing a lightweight training-free mechanism for robust diffusion-policy deployment.

## 3. Methodology

### 3.1. Problem Formulation

The goal of visual navigation is to learn a policy $\pi$ that predicts a control action $a \in \mathbb{R}^d$ and a temporal distance estimate $d \in \mathbb{R}_+$, given a sequence of past observations $\mathcal{O} = \{\boldsymbol{I}_t\}_{t=T-p}^{T}$ and a goal image $\boldsymbol{I}_g$. Each observation

$\boldsymbol{I}_t$ is encoded into a feature vector $f(\boldsymbol{I}_t)$, and the resulting sequence is fused into a context representation $\mathcal{C}$ that captures both spatial and temporal information. The policy first predicts the final action $a$ from $\mathcal{C}$ by a diffusion policy module. The same context is used to estimate $d$, reflecting how close the current state is to the goal in temporal terms. For a complete list of notation, see Sec. B.

### 3.2. Fisher Denoising Sensitivity

**Step Fisher Denoising Sensitivity**. To quantify the uncertainty of a diffusion policy's output with respect to the conditional feature $\mathcal{C}$, we define the **Fisher Denoising Sensitivity (FDS)** as follows. For a single denoising step $t$, the reconstructed action is:

$$\tilde{a}_0(\mathcal{C}, a_t, t) = \frac{a_t - \sqrt{1 - \bar{\alpha}_t}\, \epsilon_\theta(\mathcal{C}, a_t, t)}{\sqrt{\bar{\alpha}_t}}. \qquad (1)$$

We consider the input-output Jacobian:

$$J(\mathcal{C}, t) = \frac{\partial \tilde{a}_0}{\partial \mathcal{C}} = -\frac{\sqrt{1 - \bar{\alpha}_t}}{\sqrt{\bar{\alpha}_t}} \frac{\partial \epsilon_\theta(\mathcal{C}, a_t, t)}{\partial \mathcal{C}}. \qquad (2)$$

The Fisher information can be approximated as:

$$\mathcal{I}(\mathcal{C}, t) = \|J(\mathcal{C}, t)\|_F^2 = \frac{1 - \bar{\alpha}_t}{\bar{\alpha}_t} \|\nabla_{\mathcal{C}} \epsilon_\theta(\mathcal{C}, a_t, t)\|_F^2, \quad (3)$$

here, the derivative is taken with respect to the conditioning variable $\mathcal{C}$ only, because FDS is used to measure observation-conditioned sensitivity rather than to define the action-space guidance direction. This score serves as a proxy for the Fisher information and can be efficiently computed via automatic differentiation. The Step Fisher Denoising Sensitivity (Step-FDS) quantifies the local sensitivity of the predicted noise to observation perturbation:

$$\mathcal{U}_{\text{FDS}}(\mathcal{C}, t) = \mathcal{I}(\mathcal{C}, t) \qquad (4)$$

**Chain Fisher Denoising Sensitivity.** The complete denoising process involves $T$ steps. Let $a_{t-1} = G_t(\mathcal{C}, a_t)$ denote the mean update at step $t$. Unrolling the chain gives $a_0 = G_1\big(\mathcal{C}, G_2(\mathcal{C}, \ldots, G_T(\mathcal{C}, a_T))\big)$, and by the multivariate chain rule the total Jacobian from $\mathcal{C}$ to $a_0$ can be written as:

$$J(\mathcal{C}) = \frac{\partial a_0}{\partial \mathcal{C}} = \sum_{t=1}^{T} P_{t\leftarrow}\, J(\mathcal{C}, t), \quad P_{t\leftarrow} = \prod_{s=t+1}^{T} \frac{\partial G_s(\mathcal{C}, a_s)}{\partial a_s}. \qquad (5)$$

For standard samplers $G_s(\mathcal{C}, a_s) = c_s a_s - d_s\, \epsilon_\theta(\mathcal{C}, a_s, s)$, so $\partial G_s / \partial a_s = c_s I - d_s\, \partial_{a_s} \epsilon_\theta(\mathcal{C}, a_s, s)$. Since $c_s, d_s$ are known scalar factors that only rescale sensitivity, in practice, we focus on the model-dependent part and use the

normalized propagation operator:

$$\tilde{P}_{t\leftarrow} = \prod_{s=t+1}^{T} \partial_{a_s} \epsilon_\theta(\mathcal{C}, a_s, s), \qquad (6)$$

which is what we implement when computing CFDS.

The overall chain-wise FDS is then defined as the (Frobenius) Fisher norm of the total Jacobian:

$$\mathcal{U}_{\text{CFDS}}(\mathcal{C}) = \mathcal{I}(\mathcal{C}) = \big\|J(\mathcal{C})\big\|_F^2. \qquad (7)$$

For high-dimensional $J(\mathcal{C})$, we estimate this efficiently via a Hutchinson-style estimator:

$$\mathcal{I}(\mathcal{C}) \approx \frac{1}{K} \sum_{k=1}^{K} \big\|J(\mathcal{C})^\top v_k\big\|_2^2, \qquad v_k \sim \mathcal{N}(0, I). \quad (8)$$

Note that $J(\mathcal{C})$ measures how perturbations in the condition affect the final action through the reverse chain, while the reverse diffusion state itself remains $a_t$.

**Truncation FDS and Error Bound.** While the CFDS captures uncertainty over the entire reverse diffusion trajectory, its exact evaluation requires computing Jacobians at every denoising step. To address this, we introduce a truncated approximation called Truncation Fisher Denoising Sensitivity (TFDS): we accumulate Step-FDS scores only over the final $M$ denoising steps (closest to the data).

**Truncation error.** Let the full chain-wise Jacobian be $J(\mathcal{C}) = \sum_{t=1}^{T} P_{t\leftarrow} J(\mathcal{C}, t)$. Directly evaluating its Fisher norm $\|J(\mathcal{C})\|_F^2$ is expensive. In practice, we use an additive chain-FDS surrogate:

$$\bar{\mathcal{I}}(\mathcal{C}) := \sum_{t=1}^{T} \|P_{t\leftarrow} J(\mathcal{C}, t)\|_F^2, \qquad (9)$$

which accumulates the propagated sensitivity contribution from each denoising step. The truncated tail approximation keeps only the last $M$ steps closest to the data:

$$\bar{\mathcal{I}}_{\text{tail}}^{(M)} := \sum_{t=1}^{M} \|P_{t\leftarrow} J(\mathcal{C}, t)\|_F^2. \qquad (10)$$

The per-step contraction is defined as $\rho_t := 1 - \frac{1}{2}\beta_t \in (0, 1)$, where $\beta_t$ is the noise variance schedule and $\alpha_t := 1 - \beta_t$. Let $w_t := \prod_{s=t+1}^{T} \rho_s^2$ denote the cumulative propagation weight at step $t$. The relative truncation error of this additive surrogate is

$$\eta_M = \frac{\bar{\mathcal{I}}(\mathcal{C}) - \bar{\mathcal{I}}_{\text{tail}}^{(M)}}{\bar{\mathcal{I}}(\mathcal{C})}, \qquad (11)$$

and admits the following bound, detailed in Sec. E.1:

$$\eta_M \le \kappa \frac{\sum_{t=M+1}^{T} w_t}{\sum_{t=1}^{M} w_t}, \tag{12}$$

where $\kappa$ bounds the ratio of gradient magnitudes between the discarded head and the retained tail:

$$\kappa := \frac{\max_{t>M} \|J(\mathcal{C}, t)\|_F^2}{\min_{t \le M} \|J(\mathcal{C}, t)\|_F^2} \ge 1. \tag{13}$$

**Practical implication.** For typical diffusion schedules used in policy learning, such as the cosine schedule, the tail steps dominate the propagated sensitivity. For instance, with measured $\kappa \approx 1$, taking $M = 4$ out of $T = 10$ steps gives a relative surrogate error bound $\eta_4 \le 8.4\%$. Empirically, retaining only the last $M \in [4, 6]$ steps captures more than 90–95% of the full additive chain-FDS surrogate across all evaluated tasks, with negligible impact on uncertainty-based decision quality. Throughout the remainder, we denote

$$\widehat{\mathcal{U}} := \bar{\mathcal{I}}_{\text{tail}}^{(M)} \tag{14}$$

as the TFDS score.

### 3.3. Fisher-Preserving Guidance and Approximation

The previous sections quantify the uncertainty of diffusion-policy samples through Step-/Chain-FDS and use it for uncertainty-guided action blending. We now show how to steer the reverse diffusion process while preserving the Fisher sensitivity of the generated action trajectory, thereby reducing off-manifold drift when applying a task guidance loss $L$ (e.g., short-path bias or safety guidance). The complete algorithm is summarized in Algorithm 1 in Sec. I.

We distinguish two derivatives used in our formulation. FDS is computed with respect to the condition $\mathcal{C}$, measuring the sensitivity of the denoised action to observation perturbations. During guided sampling, however, $\mathcal{C}$ is fixed and the guidance loss $L$ is differentiated with respect to the current noisy action trajectory $a_t$. Thus, the Fisher-preserving projection is applied in the action-trajectory space, while preserving the FDS value defined by condition sensitivity.

**Fisher isosurface constraint.** For a fixed condition $\mathcal{C}$, we define the step-wise FDS as a function of the current diffusion state:

$$\mathcal{I}_t(a_t; \mathcal{C}) = \frac{1 - \bar{\alpha}_t}{\bar{\alpha}_t} \|\nabla_{\mathcal{C}} \varepsilon_\theta(\mathcal{C}, a_t, t)\|_F^2. \tag{15}$$

Here the derivative inside the norm is taken with respect to $\mathcal{C}$, because FDS measures sensitivity to the conditioning observation. During guided sampling, $\mathcal{C}$ remains fixed and the updated variable is $a_t$. Therefore, for each denoising

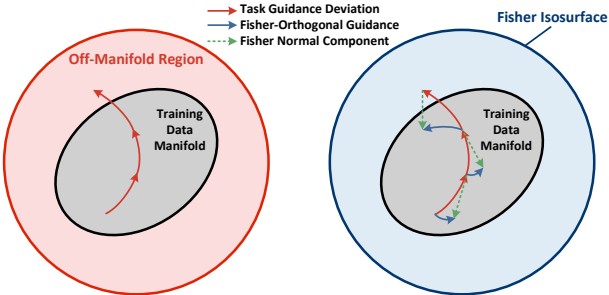

*Figure 1.* Comparison of standard task-guided diffusion (left) and Fisher-preserving guidance (right). Task guidance without constraint (red arrows) leads updates away from the training data manifold and into the off-manifold region. In contrast, Fisher-preserving guidance decomposes each update and projects it onto the Fisher isosurface (blue arrows), ensuring that the trajectory remains within the high-confidence region.

step, the Fisher isosurface is defined in the action-trajectory space:

$$S_{\kappa, t}(\mathcal{C}) = \{a_t \mid \mathcal{I}_t(a_t; \mathcal{C}) = \kappa\}. \tag{16}$$

To keep the guided update on this isosurface to first order, the update direction $\Delta_t$ should satisfy

$$g_t^\top \Delta_t = 0, \qquad g_t := \nabla_{a_t} \mathcal{I}_t(a_t; \mathcal{C}), \tag{17}$$

where $g_t$ is the Fisher normal direction in the action-trajectory space. Given a task guidance loss $L(a_t, \mathcal{C}, t)$, we compute the action-space guidance gradient

$$u_t = \nabla_{a_t} L(a_t, \mathcal{C}, t), \tag{18}$$

and project it onto the tangent space of the Fisher isosurface:

$$\Delta_t = u_t - \frac{u_t^\top g_t}{\|g_t\|^2} g_t. \tag{19}$$

The guided reverse update is then

$$a_{t-1} = \mu_t - \gamma \Delta_t. \tag{20}$$

By construction, $\Delta_t$ is orthogonal to $g_t$, so the update preserves $\mathcal{I}_t(a_t; \mathcal{C})$ up to first-order approximation, with an $O(\gamma^2)$ residual from the Taylor expansion. More derivations are provided in Sec. G.

**Low-rank approximation via Outer Product Span.** The exact evaluation of the Fisher normal vector $g_t$ in Eq. 17 involves second-order differentiation through the FDS score, which can be expensive in high-dimensional action-trajectory spaces. To reduce this cost, we exploit the low-rank structure of the denoising network head and approximate the Fisher-normal projection within an Outer Product Span (OPS) subspace.

Let $h_\theta(\mathcal{C}, a_t, t) \in \mathbb{R}^{C_h H}$ denote the latent feature before the final prediction head. We write the predicted residual noise as

$$\varepsilon_\theta(\mathcal{C}, a_t, t) \approx W h_\theta(\mathcal{C}, a_t, t), \tag{21}$$

where $W \in \mathbb{R}^{D_a \times C_h H}$ is the final linear projection and $D_a$ is the dimension of the action trajectory. The Jacobian with respect to the condition then admits the factorized form

$$\nabla_{\mathcal{C}} \varepsilon_\theta(\mathcal{C}, a_t, t) \approx W \frac{\partial h_\theta(\mathcal{C}, a_t, t)}{\partial \mathcal{C}}. \qquad (22)$$

This factorization indicates that the dominant variations of the denoising update lie in the low-dimensional subspace induced by $W$, whose rank is bounded by $C_h H$. We therefore perform the Fisher-preserving projection in this OPS subspace rather than explicitly computing the full Fisher normal direction.

To avoid evaluating $g_t = \nabla_{a_t} \mathcal{I}_t(a_t; \mathcal{C})$ exactly, we use a latent proxy $g_h$ for the Fisher normal direction in the OPS subspace. This proxy is obtained from the same backward pass used for the FDS-related Jacobian computation. Meanwhile, the condition-side gradient can be reused for Step-FDS evaluation:

$$(g_h,\ g_c) = \nabla_{(h,\mathcal{C})} \varepsilon_\theta(\mathcal{C}, a_t, t), \qquad (23)$$

where $g_h$ approximates the Fisher-normal direction in the latent OPS coordinates, and $g_c$ is reused to compute the condition sensitivity in Eq. 15. More details about OPS are provided in Sec. G.6.

**Fisher-Orthogonal Guidance Projection.** When applying an additional task loss $L$, directly using its action-space gradient may push the reverse diffusion trajectory away from the Fisher isosurface. To enforce the Fisher-preserving constraint efficiently, we restrict the update to the OPS subspace and remove the component aligned with the latent Fisher-normal direction.

Given the action-space task gradient

$$u_t = \nabla_{a_t} L(a_t, \mathcal{C}, t), \qquad (24)$$

we project it into the OPS coordinates:

$$u_h = W^\top u_t, \qquad M = W^\top W. \qquad (25)$$

Using the pullback metric $M$, we decompose $u_h$ into the Fisher-aligned and Fisher-orthogonal components:

$$u_h^{\|} = \frac{g_h^\top M u_h}{g_h^\top M g_h} g_h, \qquad u_h^{\perp} = u_h - u_h^{\|}. \qquad (26)$$

The projected update direction is then mapped back to the action-trajectory space:

$$\Delta_t = W u_h^{\perp}. \qquad (27)$$

Since $u_h^{\perp}$ is $M$-orthogonal to $g_h$, the resulting update satisfies the projected Fisher-orthogonality condition within the OPS subspace:

$$g_h^\top M u_h^{\perp} = 0. \qquad (28)$$

Equivalently, when $g_t$ is approximated by its OPS representation $W g_h$, the update approximately satisfies

$$g_t^\top \Delta_t \approx 0. \qquad (29)$$

Thus, the OPS projection provides an efficient approximation to the Fisher-preserving guidance step without explicitly computing second-order Fisher-normal derivatives.

This OPS projection reduces the per-step complexity to $O(C_h H)$, which is substantially cheaper than explicitly evaluating the full Fisher normal via second-order derivatives. Details are provided in Sec. G.6 and Sec. G.5.

### 3.4. Uncertainty-Guided Action Blending

Diffusion-based policies naturally generate a diverse set of candidate actions by sampling from a multi-modal predictive distribution. However, existing decision rules, such as randomly picking a sample or greedily selecting the most confident candidate, fail to fully exploit this diversity. In practice, these naive choices often lead to vacillating or oscillatory behaviors, where the agent hesitates between plausible options, resulting in inefficient navigation, unnecessary detours, or even collisions (Zeng et al., 2025; Xing et al., 2025). To address this, we propose an uncertainty-guided action blending strategy that fuses both sample-level stability and distributional consensus, thereby producing more robust and decisive navigation.

While the FDS provides a principled measure of input-conditioned uncertainty for each action candidate, it does not account for the typicality of a sample within the full set of generated actions. To remedy this, we introduce a cluster typicality score: for a batch of $K$ sampled actions $\{\mathbf{a}_k\}_{k=1}^K$, we cluster the actions using an unsupervised algorithm (e.g., DBSCAN), and define the typicality of each action as the normalized size of its assigned cluster:

$$C_{\text{Typ}}(\mathbf{a}_k) = \frac{|\{j \mid c_j = c_k\}|}{K} \qquad (30)$$

where $c_k$ is the cluster assignment of $\mathbf{a}_k$. This reflects how representative or mainstream an action is among the set of plausible outputs.

We then combine FDS-based uncertainty and cluster typicality into a composite confidence score:

$$C(\mathbf{a}_k) = \exp(-\eta \widehat{\mathcal{U}}(\mathbf{a}_k)) \cdot C_{\text{Typ}}(\mathbf{a}_k) \qquad (31)$$

where $\eta$ is a temperature parameter. This score jointly favors actions that are both stable under input perturbations and well-supported by the generative distribution.

The final action is computed as a weighted average:

$$a_{\text{blend}} = \frac{\sum_{k=1}^K C(\mathbf{a}_k) \mathbf{a}_k}{\sum_{k=1}^K C(\mathbf{a}_k)}. \qquad (32)$$

# 4. Experiments

This evaluation details the navigation setup (Sec. 4.1) and validates Fisher-Preserving Guidance on toy models (Sec. 4.2). We present benchmarks, ablations, and efficiency analysis (Sec. 4.3–4.5), concluding with uncertainty case studies and real-world robotic deployment (Sec. 4.6–4.7).

## 4.1. Experimental Setup

**Datasets**. To ensure a fair comparison, our method, along with all baseline approaches, was trained on a unified dataset. The training dataset encompasses a diverse set of environments and robotic platforms, incorporating data from RE-CON (Shah et al., 2021), SCAND (Karnan et al., 2022), GoStanford (Hirose et al., 2019), and SACSoN (Hirose et al., 2023). The dataset comprises sequences of consecutive image frames, each paired with corresponding positional information, providing a comprehensive training scene.

**Baselines**. We compare our approach with three state-of-the-art methods in image-based visual navigation: ViNT (Shah et al., 2023b), NoMaD (Sridhar et al., 2024) To evaluate the impact of feature representation, we selected NoMaD, which is the first approach to incorporate diffusion policies into visual navigation tasks. We integrate the proposed method with NoMaD to showcase its plug-and-play nature and the training-free enhancement it provides. We included ViNT, a regression-based model that combines self-attention and MLP for feature fusion, to compare the performance of generative models with regression-based methods in the context of visual navigation.

**Metrics**. We report three key evaluation metrics in our experiments to thoroughly assess the performance of our diffusion bridge-based visual navigation method:
*Path Length*: For tasks successfully completed, we compute the mean and variance of the path lengths to evaluate both the efficiency and consistency of the navigation.
*Collision*: The average number of collisions per trial, serving as an indicator of the navigation system's safety.
*Success Rate*: The percentage of successful trials where the robot reaches the target position within the given constraints. A trial is deemed unsuccessful if the robot does not reach the target, is due to collisions, or exceeds the time limit.

**Implementation details**. We train a base model on top of which we apply Fisher-preserving guidance. The policy uses EfficientNet-B0 as the visual backbone, followed by sparse attention and temporal shift to fuse spatio-temporal features, which are then passed to a diffusion policy to predict local actions. By default, we run 4 DDIM denoising steps (Song et al., 2020). For target image selection, we switch targets based on temporal distance following prior work (Savinov et al., 2018; Shah et al., 2023b). We optimize with Adam and a cosine annealing learning-rate schedule, using a batch size

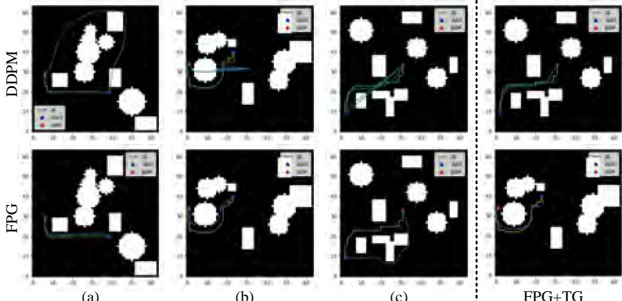

*Figure 2.* Visualization of Maze2D.

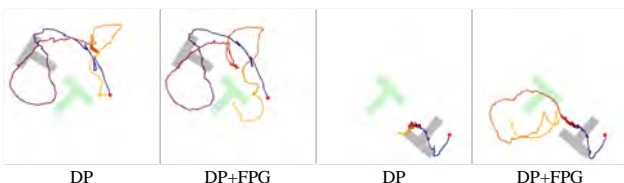

DP          DP+FPG          DP          DP+FPG

*Figure 3.* Visualization of the action path of push T tasks. The detailed process of the example is shown in Figure 11.

of 256, an initial learning rate of $10^{-4}$, and $\alpha = 10^{-4}$. The training objective matches NoMaD (Sridhar et al., 2024). The model pipeline is shown in Figure 6.

## 4.2. Toy Model Experiments

We evaluate Fisher Preserving Guidance on two complementary toy benchmarks, MAZE2D and PUSHT, to probe its effectiveness under two inference regimes: the intrinsic denoising update induced by the diffusion model and additional test-time task guidance. In both cases, naive gradient-based modifications can shift samples away from the distribution captured by the model and degrade feasibility or stability. Our goal is to test whether FPG improves this tradeoff by orthogonally decomposing the relevant gradients and adjusting the update direction to better respect the local geometry implied by the model.

Our evaluation uses task specific diffusion backbones and a shared inference interface. For MAZE2D, the baseline is a DDPM style trajectory generator designed in our framework. For a detailed description, see Sec. D. For PUSHT, we use the official pretrained weights released with Diffusion Policy (Chi et al., 2023). In all cases, we compare the original sampling procedure with the same sampler augmented with FPG and report both quantitative results and qualitative rollouts. Table 1 summarizes the main metrics across tasks, and Figures 2 and 3 visualize representative trajectories.

**MAZE2D: Safety-Constrained Planning.** For MAZE2D, we consider $G \times G$ occupancy grids with $G = 64$ and generate expert trajectories by running classical planning on inflated obstacles, followed by resampling to a fixed horizon $H$. At inference time, we optionally add a guidance term, de-

*Table 1.* Comparison of performance on Maze2D and PushT tasks. The best results are highlighted in **bold**.

| | **Maze2D** | | **PushT** | | | **Maze2D** | |
|---|---|---|---|---|---|---|---|
| Method | Colli. | Path | Score | Method | Colli. | Path |
| Baseline | 0.243 | 2.17 | 0.91 | Baseline +TG | 0.071 | 2.48 |
| FPG | **0.170** | 2.41 | **0.94** | FPG + TG | **0.016** | 2.43 |

*Table 2.* Average Performance Comparison Across All Scenarios

| Method | SR (%) ↑ | Avg. Colli. ↓ | Avg. SPL ↑ |
|---|---|---|---|
| ViNT | 53.33 | 0.611 | 0.504 |
| NoMaD | 51.11 | 0.778 | 0.478 |
| NoMaD + FPG | 60.00 | 0.644 | 0.556 |
| NoMaD + Blending | 57.78 | 0.667 | 0.521 |
| Ours | **75.55** | **0.445** | **0.653** |

*Table 3.* Comparison of Different Algorithms on GRScenes

| Alg | SR (%) | Avg. Colli. | Avg. SPL |
|---|---|---|---|
| VINT | 68.0 | 0.71 | 0.77 |
| NoMaD | 51.0 | 1.95 | 0.33 |
| Ours | **83.0** | **0.48** | **0.83** |
| Ours w/o both | 67.0 | 0.76 | 0.65 |

noted as TG, derived from a truncated signed distance field (TSDF) computed from the occupancy grid. It is visualized in Figure 9. Let $\Omega = [-1, 1]^2$ be the normalized workspace and let $s : \Omega \to \mathbb{R}$ denotes the TSDF, where larger values indicate larger clearance, and $s(\mathbf{p}) \leq 0$ indicates collision. For a waypoint trajectory $\mathbf{p}_{1:H} = (\mathbf{p}_1, \ldots, \mathbf{p}_H)$ with $\mathbf{p}_i \in \Omega$, we define the TSDF guidance cost by sampling the TSDF along the trajectory (via bilinear interpolation), denoted $\tilde{s}(\mathbf{p}_i)$, and penalizing insufficient clearance:

$$\mathcal{L}_{\text{TG}}(\mathbf{p}_{1:H}) = \sum_{i=1}^{H} \phi\left(\frac{\mu - \tilde{s}(\mathbf{p}_i)}{\tau}\right), \qquad (33)$$

where $\mu > 0$ is the desired clearance margin and $\tau > 0$ is a temperature parameter. Here $\phi : \mathbb{R} \to \mathbb{R}_+$ is a smooth, nondecreasing barrier with $\phi(z) \approx 0$ for $z \leq 0$ and increasing penalty for $z > 0$. Following the guidance injection scheme used in (Zeng et al., 2025), we apply gradient-based corrections during reverse diffusion after each denoising update. We additionally enforce endpoint constraints by inpainting, i.e., resetting the start and goal waypoints to their fixed values at every reverse step.

The results show that FPG improves safety in MAZE2D both without and with TG. Without TG, FPG reduces collision compared to the DDPM baseline while maintaining competitive path quality, as reported in Table 1. With TG enabled, the baseline already reduces collisions, but FPG further decreases collision substantially, from 0.071 to 0.016, while also slightly improving the resulting path metric from 2.48 to 2.43 in our setting. The qualitative comparisons in Figure 2 are consistent with these trends. Under the same environments and endpoints, baseline rollouts tend to exhibit unsafe behavior near obstacles or deviate from the ground truth corridor, whereas FPG produces trajectories that track feasible routes with improved clearance.

**PUSHT: Contact-Rich Manipulation.** For PUSHT, we evaluate FPG as an inference time modification on top of the official Diffusion Policy model, without introducing task specific retraining. We follow the standard closed loop receding horizon execution used by Diffusion Policy and report the task score as the primary metric (Chi et al., 2023). As shown in Table 1, adding FPG improves the score from 0.91 to 0.94. Figure 3 visualizes representative action paths and shows that DP with FPG produces more coherent pushes and avoids unstable wandering trajectories observed in the

baseline, which aligns with the interpretation that FPG mitigates harmful distribution shift during guided sampling.

### 4.3. Experiment Results

To assess robustness and versatility, we evaluate our method in two complementary simulators. First, we use CARLA (Dosovitskiy et al., 2017) with 9 routes across 3 scenes to cover diverse outdoor navigation conditions with varying route lengths and intersection structures (Fig. 10). Second, we validate transfer to constrained indoor navigation using the GRScenes dataset (Wang et al., 2024) within NVIDIA Isaac Sim, where we construct 10 routes across 5 scenes. All experiments compare our method against strong baselines, including ViNT and NoMaD, under identical protocols. Each route repeats 10 times.

Table 2 summarizes the average performance across all CARLA scenarios. The performance of each scenario is shown in Table 9. Our method achieves the best overall navigation quality, improving both task completion and path efficiency while also reducing collisions. Relative to ViNT and NoMaD, the results indicate that our policy is not only more likely to reach the goal, but also produces more efficient successful trajectories, reflecting stronger generalization across heterogeneous outdoor layouts.

The comparisons with enhanced NoMaD variants in Table 2 further clarify the contribution of our components. Incorporating Fisher-preserving guidance or uncertainty-aware action blending into NoMaD consistently improves over the vanilla backbone, supporting the plug-and-play and training-free nature of our approach. The full framework performs best overall, suggesting that these modules provide complementary benefits by jointly enhancing goal reaching reliability and navigation safety.

The indoor results in Table 3 corroborate these findings in a markedly different setting. Our approach again attains the

*Table 4.* Ablation Study on All CARLA Scenarios

| FPG | AB | SR (%) | Avg. Colli. | Avg. SPL |
|-----|-----|--------|-------------|----------|
| ✓ | ✓ | 75.55 | 0.445 | 0.653 |
| ✗ | ✓ | 57.78 | 0.622 | 0.537 |
| ✓ | ✗ | 64.44 | 0.911 | 0.576 |
| ✓ | Random | 57.78 | 0.800 | 0.518 |
| ✗ | ✗ | 48.89 | 0.700 | 0.463 |

*Table 5.* Inference efficiency comparison on visual navigation tasks (Test on RTX3060).

| Method | Step | Denoising (ms) | Total (ms) |
|--------|------|----------------|------------|
| NoMaD | 10 | 31.20 | 55.00 |
| VJP-Fisher | 10 | 85.70 | 95.32 |
| VJP-Fisher | 4 | 45.19 | 54.81 |
| Ours | 10 | 52.31 | 61.93 |
| Ours | 4 | 35.50 | 45.12 |

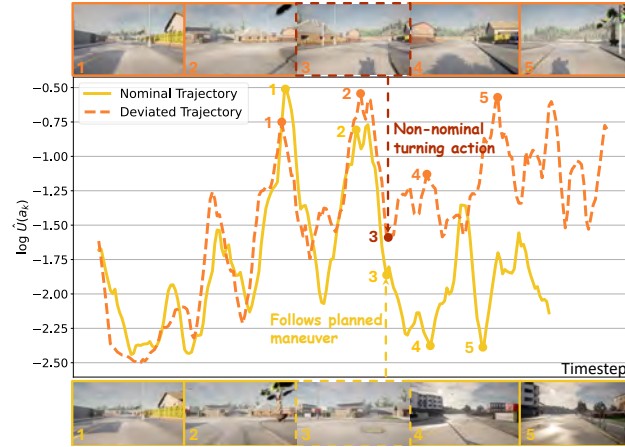

*Figure 4.* Visualization of log uncertainty for nominal and deviated trajectories. Increased uncertainty is observed when the agent encounters abnormal or erroneous observations. At intersections and other critical locations, the network demonstrates greater sensitivity to visual changes, resulting in higher uncertainty and Fisher information.

strongest success and path efficiency while incurring the fewest collisions, demonstrating effective transfer from outdoor driving-style tasks to cluttered indoor navigation. The variant without both components exhibits a clear degradation, confirming that Fisher-preserving guidance and action blending are both important for coping with perception noise and partial observability in indoor scenes. Taken together, theevaluations suggest that our framework provides a general and robust solution for visuomotor navigation across diverse environments and simulators.

### 4.4. Ablation Study

To clarify the contribution of Fisher-preserving guidance (FPG) and action blending (AB), we conduct an ablation study across all CARLA scenarios (Table 4). The full model consistently delivers the best overall performance, achieving the highest success rate and path efficiency while incurring the fewest collisions, which indicates that combining global guidance with uncertainty-aware control produces the most reliable navigation behavior.

Removing either component leads to a clear degradation. Without FPG, the policy becomes less goal-directed and tends to follow less efficient trajectories, even though AB partially stabilizes execution. In contrast, keeping FPG but disabling AB preserves coarse planning capability but increases unsafe interactions, suggesting that global guidance alone is insufficient when precise local control is required. Replacing AB with random blending further reduces performance, confirming that AB provides structured action fusion rather than incidental regularization. When both modules are absent, the model performs worst, highlighting that FPG and AB contribute complementary benefits and are jointly necessary for robust, efficient, and safe navigation.

### 4.5. Inference Efficiency Analysis

To evaluate practical deployability, we compare inference efficiency with NoMaD and VJP-Fisher on an RTX 3060 GPU (Table 5). Under the same denoising-step setting, our method incurs substantially lower latency than VJP-Fisher and remains competitive with NoMaD, indicating that the proposed guidance and blending introduce only modest overhead while improving navigation quality. We further examine a truncated denoising schedule. With fewer steps, our method achieves the lowest overall inference time among all compared approaches, outperforming both NoMaD and VJP-Fisher. Combined with the consistently strong navigation results reported earlier, these efficiency gains suggest that our framework can operate in real-time or near-real-time regimes, making it suitable for deployment in latency-sensitive visual navigation scenarios.

### 4.6. Uncertainty Analysis

To characterize policy behavior under observation perturbations, we analyze the log TFDS uncertainty $\widehat{\mathcal{U}}$ along nominal and deviated trajectories (Fig. 4). The uncertainty rises markedly when the agent experiences abnormal or inconsistent visual inputs, indicating that the policy can detect distributional shifts and correspondingly increase caution, which is critical for robust navigation under imperfect sensing. We also observe elevated uncertainty around intersections and other decision-critical regions. This pattern aligns with higher Fisher information and suggests that the policy is more sensitive to visual changes when small perceptual differences can alter the optimal action. Overall, the uncertainty signal provides an interpretable indicator of when

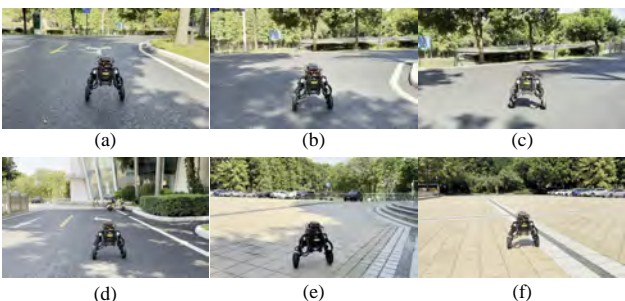

*Figure 5.* Real-world navigation. Start navigation from (a), follow the sequence as shown in the images to complete the navigation task, and (f) stop after multiple turns when near the target image.

*Table 6.* Success rates (SR) and Collision counts on real-robot settings, including average performance.

| Method | Scene 1 | | Scene 2 | | Scene 3 | | Total | |
|---|---|---|---|---|---|---|---|---|
| | SR | Coll. | SR | Coll. | SR | Coll. | SR | Coll. |
| ViNT | 8/10 | 0.2 | 7/10 | 0.3 | 7/10 | 0.4 | 22/30 | 0.37 |
| NoMaD | 7/10 | 0.3 | 7/10 | 0.5 | 5/10 | 0.9 | 19/30 | 0.57 |
| FPG | **10**/10 | **0.0** | **9**/10 | **0.1** | **9**/10 | **0.2** | **28**/30 | **0.10** |

the agent faces higher semantic ambiguity, supporting the effectiveness of our design in complex navigation scenarios.

### 4.7. Real-world Experiments

We tested the proposed FPG-OPS in a real-world environment. Experiments were conducted using a wheeled robot, Diablo, equipped with an Azure Kinect camera and an NVIDIA Jetson Orin computing platform. As shown in Figure 5, the robot successfully completed the navigation task after multiple turns to reach the destination. Quantitative experiments are shown in Sec. C.2 and Table 6.

Table 6 demonstrates that FPG improves real-world navigation reliability and safety. Compared with ViNT and NoMaD, FPG achieves higher success rates across all scenes while consistently reducing collision frequency, especially in constrained environments. These results indicate that Fisher-preserving guidance helps suppress unsafe off-manifold drift during inference, producing steadier action sequences for real-world execution.

## 5. Conclusion

**Summary** This paper presents a training-free inference framework for diffusion-based visual navigation that improves safety and robustness through Fisher-Preserving Guidance and uncertainty-aware action blending. The central idea is to control the denoising gradient and test-time guidance so that reverse diffusion updates remain close to the model's learned manifold by enforcing a Fisher-isosurface constraint while optimizing a task objective. To

make this practical, we exploit the low-rank structure of the policy head and implement the constraint efficiently with Outer-Product-Span projection. We further introduce Truncated Fisher Denoising Sensitivity as an uncertainty signal and use it to robustly fuse multiple candidate trajectories at inference time. Experiments across Maze2D with TSDF guidance, PushT with official Diffusion Policy weights, and visual navigation benchmarks including CARLA, GRScenes, and real-robot evaluation demonstrate consistent gains in safety and task performance without additional training.

**Limitations** Our method has two main limitations. First, Fisher denoising sensitivity serves as a practical proxy for local sensitivity and sample typicality, rather than a certified guarantee of physical safety. Second, our evaluation mainly focuses on RGB visual navigation, with only limited additional results on Maze2D and PushT. Future work will extend the method to broader robotic tasks, such as navigation in dynamic-obstacle scenarios and manipulation, and study its integration with stronger uncertainty estimation and safety verification mechanisms.

## Impact Statement

This work facilitates the sustainable and wide-scale deployment of diffusion policies by eliminating the energy and data costs associated with retraining. While this encourages broader adoption in diverse settings, responsible deployment remains crucial: uncertainty estimates are not calibrated safety guarantees, and practitioners must guard against overconfidence in the face of distribution shifts and dataset biases.

## Acknowledgement

This work was supported by the National Natural Science Foundation of China (U22A2095).

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

## A. Overview

This supplementary material provides comprehensive implementation details, additional experimental results, and rigorous theoretical proofs to support the main claims of our paper. The contents are organized as follows:

- **Appendix B. Notations:** A summary of the key mathematical notations and symbols used throughout the paper and appendices.

- **Appendix C. Experimental Details:** Detailed experimental settings including hyperparameters, baselines, and additional qualitative results for visual navigation tasks (including CARLA and GRScenes) and real-world robot deployment.

- **Appendix D. Additional Details for Toy Benchmarks:** Specific model architectures, data generation protocols, guidance objectives, and evaluation metrics for the Maze2D and PushT benchmarks.

- **Appendix E. Fisher Denoising Sensitivity Details:** Theoretical derivations for the Truncated Fisher Denoising Sensitivity (TFDS), including the error bound analysis for truncation and the connection to the Cramér-Rao lower bound.

- **Appendix F. Relation to Parameter Fisher Uncertainty:** A discussion clarifying the distinction between our input-space Fisher sensitivity and the conventional parameter-space Fisher information.

- **Appendix G. Properties of Fisher-Preserving Guidance:** Formal proofs regarding risk analysis, first-order loss invariance, and second-order Fisher consistency. This section also details the **Outer-Product-Span (OPS)** factorization and its efficiency in projecting updates.

- **Appendix H. Theoretical Analysis of Fisher-Preserving Dynamics:** An analysis of the optimality of FPG under external guidance and the intrinsic safety properties of Fisher-orthogonal decomposition in the absence of guidance.

- **Appendix I. Pseudocode:** A complete algorithmic description of the FPG-OPS inference loop and the Uncertainty-Guided Action Blending strategy.

## B. Notations

To facilitate reading and ensure consistency with the main text, we summarize the main notations used throughout the paper in Table 7. This appendix provides additional theoretical derivations, implementation details, and reproducibility notes to support our main results.

*Table 7.* Comprehensive Nomenclature and Symbol Definitions

| Symbol | Description |
|---|---|
| ***General Visual Navigation*** | |
| $\mathcal{O} = \{I_t\}_{t=T-p}^{T}$ | Sequence of past observation images used as input history. |
| $I_g$ | Goal image indicating the target destination. |
| $\mathcal{C}$ | Context representation encoded from observations and goal. |
| $a, a_0$ | Predicted final control action, waypoint, or action trajectory in $\mathbb{R}^d$. |
| $d$ | Predicted temporal distance to the goal. |
| $L(a_t, \mathcal{C}, t)$ | Task guidance objective function, e.g., collision avoidance or path efficiency, evaluated during reverse diffusion. |
| $\nabla_{a_t} L$ | Gradient of the task guidance loss with respect to the current noisy action state $a_t$. |
| ***Diffusion Process*** | |
| $t$ | Current denoising time step in the reverse process, $t \in \{T, \dots, 1\}$. |
| $T$ | Total number of diffusion denoising steps. |
| $a_t$ | Noisy action state at denoising step $t$. |
| $\epsilon_\theta(\mathcal{C}, a_t, t)$ | Learned noise prediction network, i.e., the diffusion denoiser. |

| | |
|---|---|
| $\mu_t$ | Mean reverse-diffusion update before applying Fisher-preserving guidance. |
| $\alpha_t, \beta_t$ | Noise schedule parameters, with $\alpha_t = 1 - \beta_t$. |
| $\bar{\alpha}_t$ | Cumulative noise schedule parameter, $\bar{\alpha}_t = \prod_{i=1}^{t} \alpha_i$. |
| $\rho_t$ | Per-step contraction factor, $\rho_t = 1 - \frac{1}{2}\beta_t$. |
| $w_t$ | Cumulative weight for chain propagation, $w_t = \prod_{s=t+1}^{T} \rho_s^2$. |

***Fisher Denoising Sensitivity (FDS)***

| | |
|---|---|
| $\tilde{a}_0(\mathcal{C}, a_t, t)$ | Reconstructed clean action estimated from the noisy action state $a_t$. |
| $J(\mathcal{C}, t)$ | Jacobian of the reconstructed action with respect to the condition, $J(\mathcal{C}, t) = \frac{\partial \tilde{a}_0}{\partial \mathcal{C}}$. |
| $\mathcal{I}(\mathcal{C}, t)$ | Step-wise Fisher-style sensitivity proxy, $\mathcal{I}(\mathcal{C}, t) = \|J(\mathcal{C}, t)\|_F^2$. |
| $\mathcal{I}_t(a_t; \mathcal{C})$ | Step-wise FDS viewed as a scalar field over the action state $a_t$ under fixed condition $\mathcal{C}$. |
| $\mathcal{U}_{\mathrm{FDS}}(\mathcal{C}, t)$ | Step Fisher Denoising Sensitivity, equivalent to $\mathcal{I}(\mathcal{C}, t)$. |
| $\mathcal{U}_{\mathrm{CFDS}}(\mathcal{C})$ | Chain Fisher Denoising Sensitivity accumulated over the reverse diffusion trajectory. |
| $\widehat{\mathcal{U}}$ or $\bar{\mathcal{I}}_{\mathrm{tail}}^{(M)}$ | Truncated FDS (TFDS): accumulated sensitivity over the final $M$ denoising steps. |
| $M$ | Truncation horizon, i.e., the tail length used for efficient sensitivity calculation. |
| $\eta_M$ | Relative truncation error bound. |
| $S_{\kappa,t}(\mathcal{C})$ | Fisher isosurface in the action-trajectory space for fixed condition $\mathcal{C}$, defined by $\mathcal{I}_t(a_t; \mathcal{C}) = \kappa$. |
| $g_t$ | Fisher normal vector in the action-trajectory space, $g_t = \nabla_{a_t}\mathcal{I}_t(a_t; \mathcal{C})$. |

***Outer Product Span (OPS) Guidance***

| | |
|---|---|
| $h_\theta(\mathcal{C}, a_t, t)$ | Latent feature representation before the final denoising prediction head. |
| $W$ | Learned linear projection matrix, or residual head, mapping latent features to the action/noise prediction space. |
| $u_t$ | Action-space task gradient, $u_t = \nabla_{a_t} L(a_t, \mathcal{C}, t)$. |
| $g_h$ | Latent OPS proxy for the Fisher normal direction. |
| $u_h$ | Task gradient projected into the OPS latent space, $u_h = W^\top u_t$. |
| $M_h$ or $M$ | Pullback metric induced by the projection head, $M_h = W^\top W$. |
| $u_h^{\parallel}$ | Component of the latent task gradient parallel to the latent Fisher normal direction. |
| $u_h^{\perp}$ | Component of the latent task gradient orthogonal to the latent Fisher normal direction. |
| $\Delta_t$ | Fisher-preserving update direction mapped back to the action-trajectory space. |
| $\gamma$ | Step size coefficient for the guidance update. |

***Uncertainty-Guided Action Blending***

| | |
|---|---|
| $K$ | Number of parallel action candidates sampled. |
| $\{a_k\}_{k=1}^{K}$ | Set of sampled action candidates. |
| $C_{\mathrm{Typ}}(a_k)$ | Cluster typicality score measuring the representativeness of sample $k$. |
| $C(a_k)$ | Composite confidence score combining TFDS and cluster typicality. |
| $\eta$ | Temperature parameter for uncertainty weighting. |
| $a_{\mathrm{blend}}$ | Final blended action computed via weighted averaging of candidates. |

# C. Experimental Details

### C.1. Experimental Settings

The core parameter settings are introduced in the implementation details of Section Experiments. Following standard practice in the field and to balance efficiency and performance, all methods use $96 \times 96$ input images and 3-frame observation histories, with AdamW as the optimizer. Training is conducted for 30 epochs, including 4 warmup epochs. The random seed is set to 0.

*Table 8.* Hyperparameter settings for training and inference.

| Parameter | Value |
|---|---|
| *Training Optimization* | |
| *Model Architecture* | |
| Vision Backbone | EfficientNet-B0 |
| Image Size | $96 \times 96$ |
| Action Horizon | 8 |
| *Fisher Inference (FPG-OPS)* | |
| Diffusion Steps | 10 |
| Guidance Scale ($\gamma$) | 0.05 |
| Candidate Samples ($K$) | 4 |
| FDS Tail Length ($M$) | 4 |
| Blending Temp ($\beta$) | 5.0 |
| Clustering Algorithm | DBSCAN ($\epsilon = 0.5$) |

## C.2. Detailed Experimental Results

Figure 7 presents a qualitative comparison of navigation trajectories generated by our Fisher-preserving diffusion policy and the NoMaD (Sridhar et al., 2024) baseline. As illustrated in subfigures (a) and (d), our method produces trajectories that closely follow the optimal path, demonstrating strong adherence to scene geometry and effective anticipation of turns and obstacles. The planned routes not only navigate around obstacles with clear margin but also exhibit smooth transitions, reflecting the robustness of our uncertainty-guided action selection. In contrast, the trajectories generated by NoMaD, shown in subfigures (b), (c), and (e), often deviate from the optimal route, especially when facing sharp corners or dense obstacles. These paths sometimes cut too close to barriers or take unnecessarily long detours, indicating a lack of risk-aware adjustment. Overall, this comparison highlights that our approach consistently delivers more reliable and efficient navigation, with a clear advantage in challenging scenarios that require precise maneuvering. The visual results further validate the effectiveness of maintaining Fisher isosurface constraints in guiding the policy towards safer and more optimal decisions.

Figure 8 presents a comparative visualization of trajectory sampling and selection between our Fisher-preserving diffusion policy and the NoMaD baseline in urban navigation scenarios. For each method, we display multiple candidate trajectories (in blue), as well as the final blended trajectory (in red) that the agent actually executes. The action selection strategy in NoMaD is implemented by choosing an arbitrary action from the entire batch without any evaluation. Our method consistently demonstrates superior multimodal exploration, producing a diverse set of feasible trajectories that account for scene geometry and potential obstacles. The final path chosen by our uncertainty-guided blending mechanism reliably steers the agent toward the goal while effectively avoiding collisions and suboptimal detours, even in ambiguous or complex street layouts. In contrast, NoMaD often exhibits less diversity among sampled paths and its selected trajectories tend to be more direct but also more susceptible to risk, frequently resulting in less robust navigation and, in some cases, increased

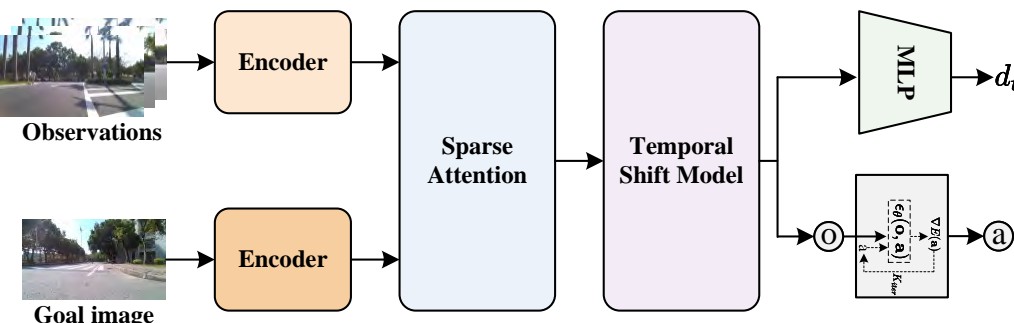

*Figure 6.* Pipeline of the base model of FPG, modified from (Ren et al., 2026).

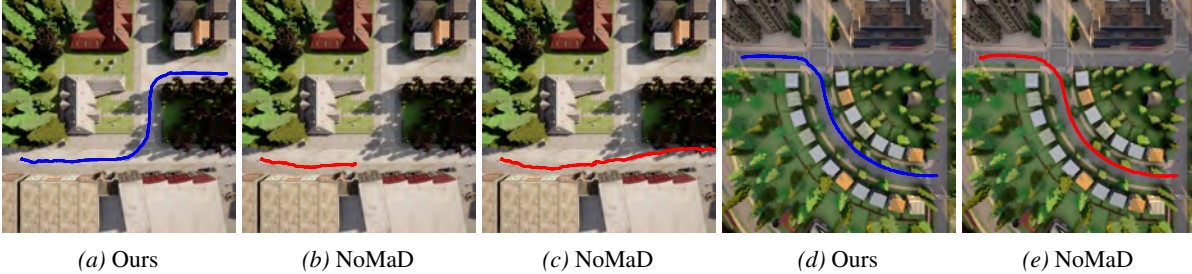

*(a)* Ours     *(b)* NoMaD     *(c)* NoMaD     *(d)* Ours     *(e)* NoMaD

*Figure 7.* Qualitative comparison of navigation trajectories. (a, d) Trajectories generated by our Fisher-preserving diffusion policy. (b, c, e) Trajectories generated by NoMaD. Our method consistently produces more accurate and efficient paths, especially when navigating around obstacles or sharp turns, while NoMaD trajectories are less optimal and sometimes exhibit significant deviation.

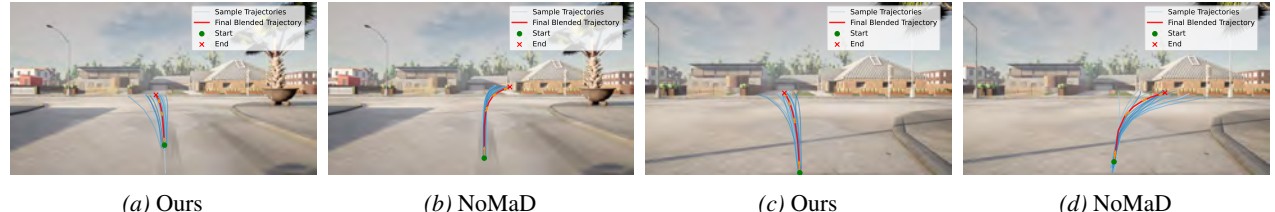

*(a)* Ours     *(b)* NoMaD     *(c)* Ours     *(d)* NoMaD

*Figure 8.* Visualization of sampled and final blended trajectories in urban navigation scenarios. (a, c) Our method generates diverse candidate trajectories (blue) and reliably blends them into optimal paths (red) towards the goal, demonstrating robustness to uncertainty and environmental variability. (b, d) NoMaD trajectories are less adaptive, often leading to suboptimal or riskier paths. Green dots indicate the starting position; red crosses indicate the goal.

likelihood of failure to reach the goal. These visual results underscore the practical benefits of Fisher-preserving guidance for safe and efficient decision making in visually complex real-world environments.

In the real-robot evaluation, we test three indoor scenes with identical sensing, control, and trial protocol across methods. We report success rate (SR) as the number of successful episodes over the total number of trials, and we report collision counts (Coll.) as the average number of collisions per episode in each scene and averaged across all scenes. This setting is intentionally challenging due to actuation noise, perception errors, and scene-specific distribution shift, so improvements here are indicative of practical robustness rather than purely simulation gains.

Table 6 shows that FPG provides a consistent improvement in safety while also increasing reliability. Compared to ViNT and NoMaD, FPG achieves higher success across all scenes and, more importantly, substantially lowers collision frequency in every scene. The gap is most pronounced in the more constrained scenes, where baseline methods tend to exhibit occasional unsafe interactions even when they reach the goal, while FPG maintains low-collision behavior without sacrificing completion. Overall, these results suggest that the Fisher-preserving update reduces harmful off-manifold drift during inference, leading to steadier action sequences that transfer better to real-world execution.

*Table 9.* Comparison of Different Methods Across Scenarios

| Method | Scenario 1 | | | Scenario 2 | | | Scenario 3 | | |
|---|---|---|---|---|---|---|---|---|---|
| | SR (%) | Avg. Colli. | Avg. SPL | SR (%) | Avg. Colli. | Avg. SPL | SR (%) | Avg. Colli. | Avg. SPL |
| ViNT | 66.67 | 0.533 | 0.649 | 33.33 | 0.833 | 0.309 | 60.00 | 0.467 | 0.554 |
| NoMaD | 60.00 | 0.533 | 0.594 | 53.33 | 0.733 | 0.462 | 40.00 | 1.067 | 0.377 |
| NoMaD + FPG | 66.67 | 0.500 | 0.616 | 60.00 | 0.700 | 0.529 | 53.33 | 0.733 | 0.523 |
| NoMaD + Blending | 66.67 | **0.467** | 0.603 | 60.00 | **0.667** | 0.498 | 46.67 | 0.867 | 0.463 |
| Ours | **73.33** | **0.467** | **0.644** | **73.33** | **0.667** | **0.625** | **80.00** | **0.200** | **0.689** |

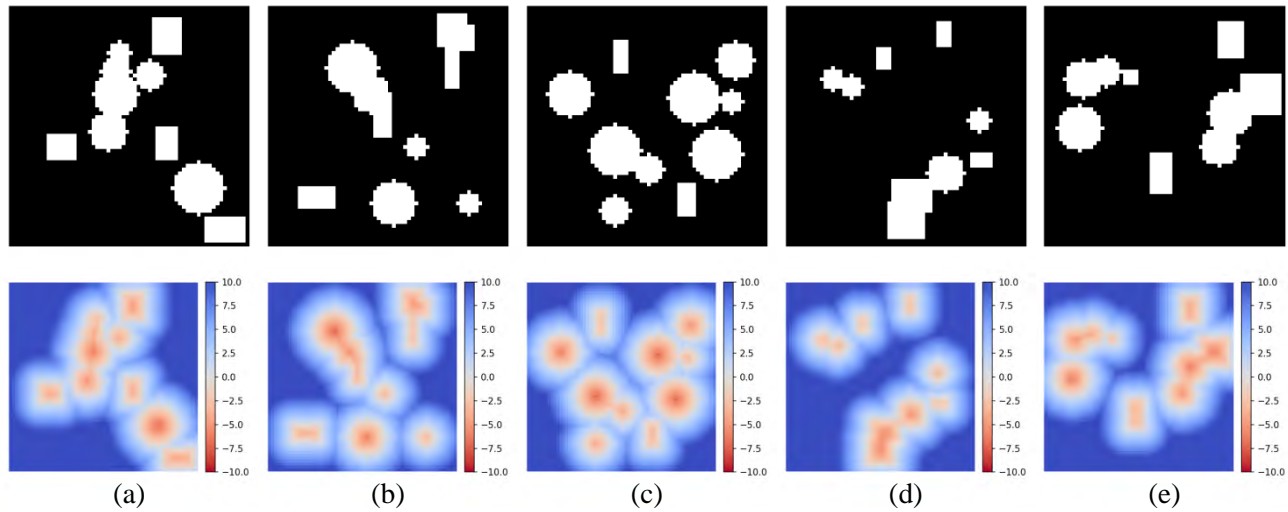

*Figure 9.* Visualization of the obstacle grid map and corresponding TSDF map.

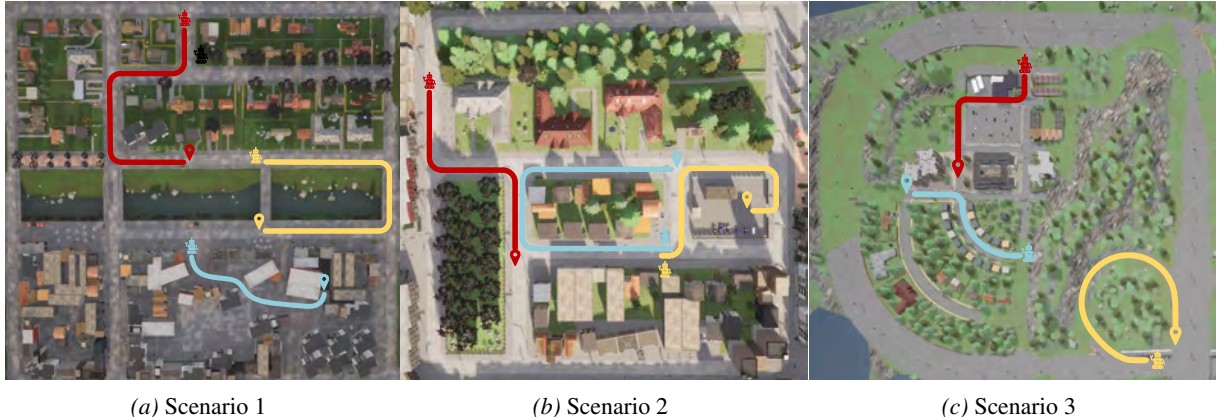

*(a)* Scenario 1  *(b)* Scenario 2  *(c)* Scenario 3

*Figure 10.* Top-down views of the nine experimental tasks. Each scenario features distinct urban or suburban layouts, with predefined start and goal positions indicated by colored markers and corresponding trajectories. The red, blue, and yellow paths represent different agent routes designed to evaluate performance under varying environmental complexity.

## D. Additional Details for Toy Benchmarks

### D.1. Model Architectures

**MAZE2D policy.** For MAZE2D, we use a conditional DDPM-style policy that predicts noise for a fixed-horizon waypoint sequence. The model consists of (i) a spatial condition encoder that extracts both a global conditioning vector and a set of spatial tokens from the occupancy map using a ResNet-18 backbone, and (ii) a 1D U-Net noise predictor over the waypoint horizon. Conditioning is injected through FiLM-style modulation using the global vector, and cross-attention from the trajectory features to the map tokens to preserve spatial reasoning. The last hidden feature map before the final $1 \times 1$ projection is cached as a latent representation for Fisher-related computations.

**PUSHT policy.** For PUSHT, we follow the standard Diffusion Policy setup and use the official pretrained model weights. The policy predicts an action chunk of fixed horizon conditioned on a short observation history, and is executed in a receding-horizon manner by applying the first action and replanning at the next timestep. We do not retrain the model when evaluating inference-time guidance variants.

## D.2. Datasets and Task Setup

**MAZE2D data generation.**   We generate $G \times G$ obstacle fields with $G = 64$ and sample reachable start-goal pairs. Expert trajectories are computed by classical planning on inflated obstacles to enforce a safety margin, and then resampled to a fixed horizon $H$ and normalized to the workspace $\Omega = [-1, 1]^2$. We adopt an inpainting formulation during diffusion sampling by fixing the start and goal waypoints at every reverse step using a binary mask and corresponding target values.

**PUSHT evaluation protocol.**   We use the standard PUSHT dataset and evaluation procedure from Diffusion Policy. In each episode, the policy receives a short history of observations and generates an action sequence; the environment executes the first action and the policy replans at the next timestep. We report the task score used by the benchmark and compute statistics over a fixed set of evaluation seeds.

## D.3. Guidance Objectives and Implementation

**TSDF task guidance for MAZE2D.**   When enabled, we apply an additional test-time guidance term (TG) based on a truncated signed distance field (TSDF) computed from the occupancy grid. Let $s : \Omega \to \mathbb{R}$ denote the TSDF where larger values indicate larger clearance. For a waypoint trajectory $\mathbf{p}_{1:H}$, the guidance cost penalizes low clearance along the trajectory,

$$\mathcal{L}_{\mathrm{TG}}(\mathbf{p}_{1:H}) = \sum_{i=1}^{H} \phi\left(\frac{\mu - \tilde{s}(\mathbf{p}_i)}{\tau}\right), \tag{34}$$

where $\tilde{s}(\mathbf{p}_i)$ denotes bilinear interpolation of the discrete TSDF grid at waypoint $\mathbf{p}_i$, $\mu$ is a clearance margin, and $\tau$ is a temperature. We use a smooth nondecreasing barrier $\phi$ and apply TG inside reverse diffusion after each denoising step.

**Receding-horizon inference for PUSHT.**   For PUSHT, we evaluate guidance only at inference time using the same receding-horizon loop as the baseline. All methods share the same pretrained checkpoint and differ only in the sampling rule, enabling a direct comparison of inference-time modifications.

## D.4. Metrics and Reporting

**MAZE2D.**   We report collision rate, success rate (final waypoint within a tolerance of the goal), and a path-quality metric based on trajectory length in normalized coordinates. Collisions are detected by sampling the distance/TSDF field along the trajectory and checking whether clearance falls below a fixed threshold.

**PUSHT.**   We report the benchmark task score for each episode and aggregate results across evaluation seeds. When comparing sampling variants, we keep the number of diffusion steps and the number of sampled candidates fixed.

# E. Fisher Denoising Sensitivity Details

## E.1. Derivation of Truncated FDS Error Bound

**Definitions.**   We analyze the reverse diffusion process proceeding from $t = T$ (noise) down to $t = 1$ (data). For each denoising step, let

$$J(\mathcal{C}, t) = \frac{\partial \tilde{a}_0}{\partial \mathcal{C}}$$

denote the step-wise condition-side Jacobian used by FDS, and let

$$w_t := \|P_{t\leftarrow}\|^2 = \prod_{s=t+1}^{T} \rho_s^2$$

be the cumulative propagation weight from step $t$ to the final action.

Consistent with the main text, we analyze the additive chain-FDS surrogate

$$\bar{\mathcal{I}}(\mathcal{C}) := \sum_{t=1}^{T} w_t \|J(\mathcal{C}, t)\|_F^2. \tag{35}$$

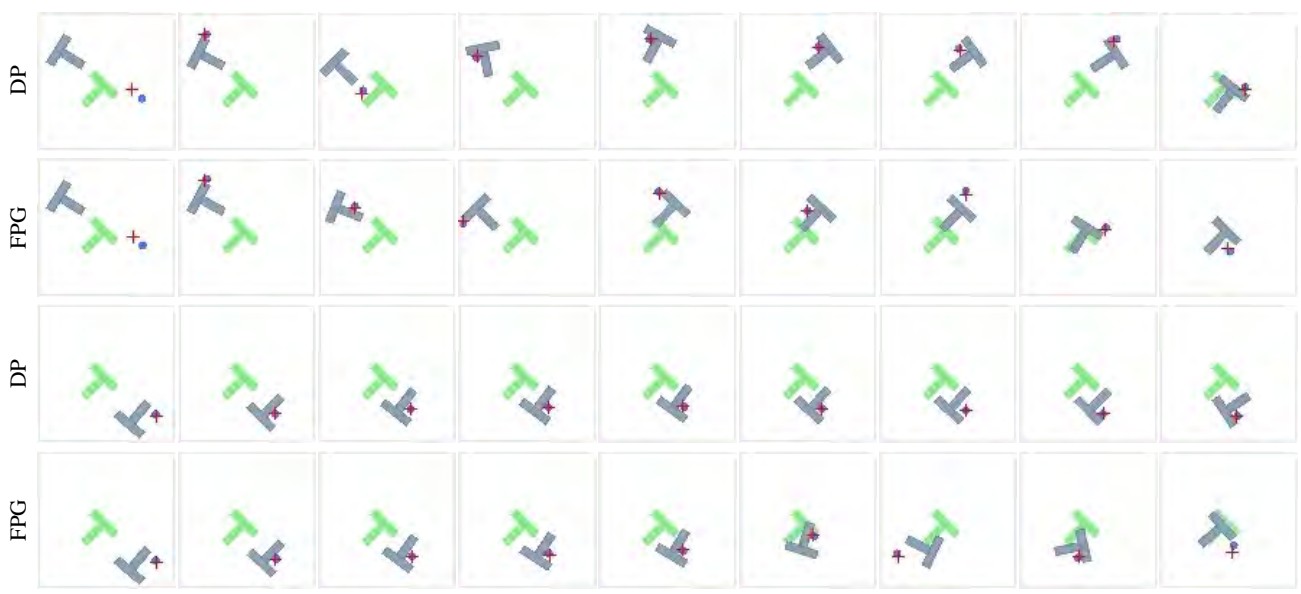

*Figure 11.* Visualization of the push T task case process.

The retained **Tail** consists of the final $M$ denoising steps, $t \in \{1, \ldots, M\}$, while the discarded **Head** consists of the earlier noisy steps, $t \in \{M+1, \ldots, T\}$. We define

$$\bar{\mathcal{I}}_{\text{tail}}^{(M)}(\mathcal{C}) := \sum_{t=1}^{M} w_t \|J(\mathcal{C}, t)\|_F^2, \tag{36}$$

$$\bar{\mathcal{I}}_{\text{head}}^{(M)}(\mathcal{C}) := \sum_{t=M+1}^{T} w_t \|J(\mathcal{C}, t)\|_F^2. \tag{37}$$

Thus,

$$\bar{\mathcal{I}}(\mathcal{C}) = \bar{\mathcal{I}}_{\text{tail}}^{(M)}(\mathcal{C}) + \bar{\mathcal{I}}_{\text{head}}^{(M)}(\mathcal{C}). \tag{38}$$

**Gradient norm bound.** Let $\kappa \geq 1$ bound the ratio between the largest step-wise Jacobian norm in the discarded head and the smallest one in the retained tail:

$$\kappa := \frac{\max_{t>M} \|J(\mathcal{C}, t)\|_F^2}{\min_{t \leq M} \|J(\mathcal{C}, t)\|_F^2}. \tag{39}$$

Equivalently, if we denote

$$g_{\min} := \min_{t \leq M} \|J(\mathcal{C}, t)\|_F^2, \tag{40}$$

then for all $t > M$,

$$\|J(\mathcal{C}, t)\|_F^2 \leq \kappa g_{\min}, \tag{41}$$

and for all $t \leq M$,

$$\|J(\mathcal{C}, t)\|_F^2 \geq g_{\min}. \tag{42}$$

**Error bound derivation.** The relative truncation error of the additive chain-FDS surrogate is

$$\eta_M := \frac{\bar{\mathcal{I}}(\mathcal{C}) - \bar{\mathcal{I}}_{\text{tail}}^{(M)}(\mathcal{C})}{\bar{\mathcal{I}}(\mathcal{C})} = \frac{\bar{\mathcal{I}}_{\text{head}}^{(M)}(\mathcal{C})}{\bar{\mathcal{I}}_{\text{head}}^{(M)}(\mathcal{C}) + \bar{\mathcal{I}}_{\text{tail}}^{(M)}(\mathcal{C})}. \tag{43}$$

Since both terms are nonnegative, we have

$$\eta_M \leq \frac{\bar{\mathcal{I}}_{\text{head}}^{(M)}(\mathcal{C})}{\bar{\mathcal{I}}_{\text{tail}}^{(M)}(\mathcal{C})}. \tag{44}$$

We now upper-bound the head and lower-bound the tail. For the discarded head,

$$\bar{\mathcal{I}}_{\text{head}}^{(M)}(\mathcal{C}) = \sum_{t=M+1}^{T} w_t \|J(\mathcal{C}, t)\|_F^2$$
$$\leq \kappa g_{\min} \sum_{t=M+1}^{T} w_t. \tag{45}$$

For the retained tail,

$$\bar{\mathcal{I}}_{\text{tail}}^{(M)}(\mathcal{C}) = \sum_{t=1}^{M} w_t \|J(\mathcal{C}, t)\|_F^2$$
$$\geq g_{\min} \sum_{t=1}^{M} w_t. \tag{46}$$

Combining the two inequalities gives

$$\eta_M \leq \frac{\kappa g_{\min} \sum_{t=M+1}^{T} w_t}{g_{\min} \sum_{t=1}^{M} w_t} = \kappa \frac{\sum_{t=M+1}^{T} w_t}{\sum_{t=1}^{M} w_t}. \tag{47}$$

### E.2. Cramér–Rao Lower Bound and Fisher Denoising Sensitivity

The Cramér–Rao lower bound (CRLB) provides a classical connection between Fisher information and the variance of unbiased estimators. In our setting, however, Fisher Denoising Sensitivity (FDS) is not used as a certified statistical Fisher information matrix. Instead, it serves as a Fisher-style local sensitivity proxy for the mapping from the conditioning representation $\mathcal{C}$ to the generated action $a_0$.

Specifically, we define the FDS score as

$$\mathcal{U}_{\text{FDS}}(\mathcal{C}) = \left\| \frac{\partial a_0}{\partial \mathcal{C}} \right\|_F^2. \tag{48}$$

This quantity measures how strongly the generated action changes under infinitesimal perturbations of the conditioning observation. A larger FDS value therefore indicates higher local sensitivity and lower action stability with respect to observation perturbations.

**Connection to the CRLB.** Under an implicit local noise model in which perturbations of $\mathcal{C}$ induce variability in the generated action, the FDS score can be viewed as a Fisher-style sensitivity measure related to estimator variance. This provides an intuitive connection to the Cramér–Rao framework: regions with high local sensitivity are more likely to produce unstable action estimates, while regions with low sensitivity correspond to more stable predictions.

**Extension to Truncated/Chain FDS.** TFDS and CFDS extend this local sensitivity view to the reverse diffusion process. CFDS accumulates sensitivity over the full denoising chain, while TFDS provides an efficient tail approximation using the final $M$ denoising steps:

$$\widehat{\mathcal{U}} = \bar{\mathcal{I}}_{\text{tail}}^{(M)}. \tag{49}$$

Thus, TFDS is used as a practical uncertainty proxy for sample selection, action blending, and risk estimation. We emphasize that this interpretation provides a useful Fisher-style motivation, rather than a certified lower bound on estimator variance.

## F. Relation to Parameter Fisher Uncertainty

**Two Fisher Views.** Most Bayesian– or variational–uncertainty works (e.g., *Bayes by Backprop, KFAC*) quantify *parameter uncertainty*: they study how the predictive distribution $p_\theta(y | \mathcal{C})$ varies under infinitesimal *parameter* perturbations $\delta\theta$. The resulting metric is the **parameter-Fisher information**

$$\mathcal{I}_\theta^{\text{param}} = \mathbb{E}_{y \sim p_\theta} \left[ \nabla_\theta \log p_\theta(y \mid \mathcal{C}) \, \nabla_\theta \log p_\theta(y \mid \mathcal{C})^\top \right]. \tag{50}$$

Because a larger $\mathcal{I}_\theta^{\text{param}}$ implies a *tighter* Cramér–Rao bound, the classical literature interprets "*Fisher* $\uparrow$ $\Leftrightarrow$ *uncertainty* $\downarrow$".

**Input-Fisher for Decision Robustness.** In contrast, our work focuses on *input robustness*: given a fixed, trained policy $\theta^\star$, we ask how sensitive the generated action $a_0$ is to infinitesimal *observation* perturbations $\delta\mathcal{C}$:

$$J(\mathcal{C}, t) = \frac{\partial \tilde{a}_0}{\partial \mathcal{C}}, \qquad \mathcal{U}_{\text{Step-FDS}}(\mathcal{C}, t) = \|J(\mathcal{C}, t)\|_2^2. \tag{51}$$

This *input-Fisher* (FDS) can be interpreted as a Fisher-style local sensitivity proxy where the perturbed variable is $\mathcal{C}$ rather than the model parameter $\theta$. Here, a *larger* value means *higher* local sensitivity of the action to nuisance changes in the perceptual input, hence **"FDS $\uparrow$ $\Rightarrow$ uncertainty $\uparrow$"**.

## G. Properties of Fisher-Preserving Guidance

### G.1. Why Fisher-Preserving Guidance Reduces Risk

We summarize the main advantages of maintaining a constant Fisher sensitivity during guided reverse diffusion. Unlike unconstrained guidance, which may push the noisy action state $a_t$ toward regions with unstable sensitivity, Fisher-preserving guidance constrains the update to the tangent space of a Fisher isosurface. This helps reduce off-manifold drift while still allowing task-oriented improvement.

| Risk | Cause if Fisher sensitivity is ignored | Mitigation via Fisher-preserving guidance |
| --- | --- | --- |
| Off-manifold drift | Large guidance strength $\gamma$ may push the action state $a_t$ along the Fisher-normal direction, moving it into regions with abnormal sensitivity. | Tangent-space updates preserve $\mathcal{I}_t(a_t; \mathcal{C})$ up to first order, reducing harmful drift. |
| Gradient explosion/collapse | Excessive Fisher sensitivity can amplify small perturbations, while vanishing sensitivity may indicate degenerate or uninformative samples. | Maintaining a stable Fisher radius keeps the sample within a more reliable sensitivity regime. |
| Loss–safety trade-off | A stronger task gradient may reduce the guidance loss but also increase Fisher drift. | Projection removes the Fisher-aligned component of the task gradient and keeps only the Fisher-orthogonal descent direction. |

*Table 10.* Risk analysis of unconstrained guidance and Fisher-preserving guidance.

### G.2. Fisher-Preserving Step as a Constrained Descent Direction

For a fixed condition $\mathcal{C}$, we view the step-wise FDS as a scalar field over the current noisy action state:

$$\mathcal{I}_t(a_t; \mathcal{C}) = \frac{1 - \bar{\alpha}_t}{\bar{\alpha}_t} \|\nabla_{\mathcal{C}} \epsilon_\theta(\mathcal{C}, a_t, t)\|_F^2. \tag{52}$$

Here, the derivative inside the norm is taken with respect to the condition $\mathcal{C}$, because FDS measures observation-conditioned sensitivity. During guided sampling, however, $\mathcal{C}$ is fixed and the updated variable is $a_t$.

The Fisher normal direction in the action-trajectory space is

$$g_t := \nabla_{a_t} \mathcal{I}_t(a_t; \mathcal{C}). \tag{53}$$

Let the task guidance gradient be

$$u_t := \nabla_{a_t} L(a_t, \mathcal{C}, t). \tag{54}$$

The Fisher-preserving update removes the component of $u_t$ aligned with $g_t$:

$$u_t^\perp = u_t - \frac{u_t^\top g_t}{\|g_t\|^2} g_t. \tag{55}$$

The corresponding update is

$$\Delta_t = -\gamma u_t^{\perp}. \tag{56}$$

By construction,

$$g_t^{\top} \Delta_t = 0, \tag{57}$$

so $\Delta_t$ lies in the tangent space of the Fisher isosurface.

**First-order loss reduction.** Using the first-order Taylor expansion of the task loss around $a_t$, we have

$$L(a_t + \Delta_t, \mathcal{C}, t) = L(a_t, \mathcal{C}, t) + u_t^{\top} \Delta_t + O(\|\Delta_t\|^2). \tag{58}$$

Substituting $\Delta_t = -\gamma u_t^{\perp}$ gives

$$L(a_t + \Delta_t, \mathcal{C}, t) = L(a_t, \mathcal{C}, t) - \gamma \|u_t^{\perp}\|^2 + O(\gamma^2). \tag{59}$$

Thus, Fisher-preserving guidance still provides a valid first-order descent direction whenever the task gradient has a nonzero component tangent to the Fisher isosurface. The removed component is precisely the part that would change the Fisher sensitivity to first order.

**Fisher drift suppression.** Expanding the FDS scalar field around $a_t$, we obtain

$$\mathcal{I}_t(a_t + \Delta_t; \mathcal{C}) = \mathcal{I}_t(a_t; \mathcal{C}) + g_t^{\top} \Delta_t + \frac{1}{2} \Delta_t^{\top} H_{\mathcal{I}} \Delta_t + o(\|\Delta_t\|^2), \tag{60}$$

where $H_{\mathcal{I}}$ is the Hessian of $\mathcal{I}_t$ with respect to $a_t$. Since $g_t^{\top} \Delta_t = 0$, the first-order Fisher drift vanishes:

$$\mathcal{I}_t(a_t + \Delta_t; \mathcal{C}) - \mathcal{I}_t(a_t; \mathcal{C}) = O(\gamma^2). \tag{61}$$

In contrast, an unconstrained update $\Delta_{\text{raw}} = -\gamma u_t$ generally yields

$$\mathcal{I}_t(a_t + \Delta_{\text{raw}}; \mathcal{C}) - \mathcal{I}_t(a_t; \mathcal{C}) = -\gamma g_t^{\top} u_t + O(\gamma^2), \tag{62}$$

which can introduce first-order Fisher drift.

**Composite risk interpretation.** The above analysis shows that Fisher-preserving guidance trades the Fisher-aligned component of the task gradient for improved stability. While the raw gradient may decrease the task loss faster in the first order, it can also change the Fisher sensitivity at $O(\gamma)$. In contrast, FPG decreases the task loss along the feasible tangent direction and suppresses Fisher drift to $O(\gamma^2)$. This makes the update preferable when the task objective must be optimized without moving the sample into regions of unstable or atypical sensitivity.

### G.3. Fisher Isosurface Constraint: First-Order Invariance

**Lemma 1.** Let $\mathcal{I}_t(a_t; \mathcal{C})$ be differentiable with respect to $a_t$, with fixed condition $\mathcal{C}$. For a small update $\Delta_t$, we have

$$\mathcal{I}_t(a_t + \Delta_t; \mathcal{C}) = \mathcal{I}_t(a_t; \mathcal{C}) + \nabla_{a_t} \mathcal{I}_t(a_t; \mathcal{C})^{\top} \Delta_t + o(\|\Delta_t\|). \tag{63}$$

Therefore, $\mathcal{I}_t$ is preserved up to first order if

$$\nabla_{a_t} \mathcal{I}_t(a_t; \mathcal{C})^{\top} \Delta_t = 0. \tag{64}$$

### G.4. Orthogonal Projection and Loss Guarantee

**Theorem 1.** Consider minimizing a differentiable guidance loss $L(a_t, \mathcal{C}, t)$ under the first-order Fisher-preserving constraint

$$g_t^{\top} \Delta_t = 0, \qquad g_t = \nabla_{a_t} \mathcal{I}_t(a_t; \mathcal{C}). \tag{65}$$

The first-order constrained descent direction is obtained by projecting the action-space task gradient onto the tangent space of the Fisher isosurface:

$$\Delta_t^* = -\gamma \left[ u_t - \frac{u_t^{\top} g_t}{\|g_t\|^2} g_t \right], \qquad u_t = \nabla_{a_t} L(a_t, \mathcal{C}, t). \tag{66}$$

**Proof.** Using the first-order expansion

$$L(a_t + \Delta_t, \mathcal{C}, t) \approx L(a_t, \mathcal{C}, t) + u_t^\top \Delta_t, \tag{67}$$

we seek a descent direction satisfying $g_t^\top \Delta_t = 0$. This is equivalent to removing from $u_t$ its component parallel to $g_t$. The projected gradient is

$$u_t^\perp = u_t - \frac{u_t^\top g_t}{\|g_t\|^2} g_t. \tag{68}$$

Taking a step along $-u_t^\perp$ gives

$$\Delta_t^* = -\gamma u_t^\perp = -\gamma \left[ u_t - \frac{u_t^\top g_t}{\|g_t\|^2} g_t \right]. \tag{69}$$

Moreover,

$$g_t^\top \Delta_t^* = -\gamma \left[ g_t^\top u_t - \frac{u_t^\top g_t}{\|g_t\|^2} g_t^\top g_t \right] = 0. \tag{70}$$

Thus, the update is tangent to the Fisher isosurface and is the steepest first-order descent direction within this tangent space.

**Corollary.** Let $a_{t-1} = a_t + \Delta_t^*$. Then

$$L(a_{t-1}, \mathcal{C}, t) = L(a_t, \mathcal{C}, t) - \gamma \|u_t^\perp\|^2 + O(\gamma^2). \tag{71}$$

Therefore, FPG decreases the task loss along the feasible Fisher-preserving direction while removing the first-order Fisher-changing component.

### G.5. Properties of Fisher-Preserving Update

**Lemma 2.** Under the Fisher-preserving update $\Delta_t^*$, the change in FDS satisfies

$$\mathcal{I}_t(a_t + \Delta_t^*; \mathcal{C}) = \mathcal{I}_t(a_t; \mathcal{C}) + O(\|\Delta_t^*\|^2). \tag{72}$$

**Proof.** By second-order Taylor expansion with respect to $a_t$,

$$\mathcal{I}_t(a_t + \Delta_t^*; \mathcal{C}) = \mathcal{I}_t(a_t; \mathcal{C}) + g_t^\top \Delta_t^* + \frac{1}{2}(\Delta_t^*)^\top H_\mathcal{I} \Delta_t^* + o(\|\Delta_t^*\|^2). \tag{73}$$

Since $g_t^\top \Delta_t^* = 0$, the first-order term vanishes, leaving only second-order and higher-order terms.

### G.6. Outer-Product-Span (OPS) Factorization and Projection

**Lemma 3.** Let $h_\theta(\mathcal{C}, a_t, t) \in \mathbb{R}^{C_h H}$ be the latent feature before the final denoising prediction head, and approximate the predicted residual noise as

$$\epsilon_\theta(\mathcal{C}, a_t, t) \approx W h_\theta(\mathcal{C}, a_t, t), \tag{74}$$

where $W \in \mathbb{R}^{D_a \times C_h H}$. Then the condition-side Jacobian used by FDS admits the factorized form

$$\nabla_\mathcal{C} \epsilon_\theta(\mathcal{C}, a_t, t) \approx W \frac{\partial h_\theta(\mathcal{C}, a_t, t)}{\partial \mathcal{C}}. \tag{75}$$

This indicates that the dominant variations induced by the prediction head lie in a subspace whose rank is bounded by $C_h H$, enabling an efficient OPS approximation.

**OPS projection.** Given the action-space task gradient

$$u_t = \nabla_{a_t} L(a_t, \mathcal{C}, t), \tag{76}$$

we project it into OPS coordinates:

$$u_h = W^\top u_t, \qquad M_h = W^\top W. \tag{77}$$

Let $g_h$ denote the latent OPS proxy of the Fisher normal direction. We remove the component of $u_h$ aligned with $g_h$ under the pullback metric $M_h$:

$$u_h^{\parallel} = \frac{g_h^{\top} M_h u_h}{g_h^{\top} M_h g_h} g_h, \qquad u_h^{\perp} = u_h - u_h^{\parallel}. \tag{78}$$

The projected update direction is mapped back to action space as

$$\Delta_t = W u_h^{\perp}. \tag{79}$$

Since $u_h^{\perp}$ is $M_h$-orthogonal to $g_h$, we have

$$g_h^{\top} M_h u_h^{\perp} = 0. \tag{80}$$

When the action-space Fisher normal $g_t$ is approximated by its OPS representation $W g_h$, this gives the projected Fisher-orthogonality condition

$$g_t^{\top} \Delta_t \approx 0. \tag{81}$$

Thus, OPS provides an efficient approximation to the Fisher-preserving projection without explicitly computing the full second-order Fisher normal.

**Complexity comparison.** The OPS projection requires $O(C_h H)$ operations per denoising step, whereas explicit computation of the full Fisher normal can require substantially higher cost due to second-order differentiation in the full action-trajectory space.

**Practical implication.** By preserving the FDS value during guidance up to first order, FPG keeps the reverse diffusion trajectory within a stable sensitivity regime while still allowing task-oriented action refinement. This provides a principled inference-time mechanism for improving robustness without retraining the diffusion policy.

### G.7. Additional Notes

**Cramér–Rao Bound and FDS.** The classical Cramér–Rao lower bound connects Fisher information to estimator variance. In our setting, FDS should be interpreted as a Fisher-style local sensitivity proxy for the mapping from condition $\mathcal{C}$ to the generated action, rather than as a certified statistical lower bound.

**TFDS and Multi-Modal Blending.** TFDS enables efficient sample-level sensitivity estimation over the final denoising steps. Combining TFDS with cluster typicality allows the policy to favor samples that are both locally stable under observation perturbations and representative of the generated action distribution.

## H. Theoretical Analysis of Fisher-Preserving Dynamics

### H.1. Preliminaries and Definitions

Let $a_t \in \mathbb{R}^d$ denote the **latent action state** at a given diffusion timestep $t$, and let $\mathcal{C}$ denote the fixed conditioning observation. Let $\epsilon_{\theta}(\mathcal{C}, a_t, t)$ be the denoising model.

Consistent with the main text, we define the **Step-wise Fisher Sensitivity** as a scalar field over the action state under fixed condition $\mathcal{C}$:

$$\mathcal{I}_t(a_t; \mathcal{C}) = \frac{1 - \bar{\alpha}_t}{\bar{\alpha}_t} \|\nabla_{\mathcal{C}} \epsilon_{\theta}(\mathcal{C}, a_t, t)\|_F^2. \tag{82}$$

Here, the derivative inside the norm is taken with respect to $\mathcal{C}$, because FDS measures sensitivity to observation perturbations. The gradient of this sensitivity field with respect to the action state, denoted as the **Fisher Normal**, is:

$$g_t(a_t) := \nabla_{a_t} \mathcal{I}_t(a_t; \mathcal{C}) \in \mathbb{R}^d. \tag{83}$$

We define the **Fisher Isosurface** $S_{\kappa, t}(\mathcal{C})$ as the level set of action states with constant sensitivity:

$$S_{\kappa, t}(\mathcal{C}) = \{a_t \in \mathbb{R}^d \mid \mathcal{I}_t(a_t; \mathcal{C}) = \kappa\}. \tag{84}$$

For a small update vector $\Delta_t$ applied to the state $a_t$, the local manifold constraint requires $\Delta_t$ to be tangent to $S_{\kappa, t}(\mathcal{C})$, satisfying the orthogonality condition:

$$g_t(a_t)^{\top} \Delta_t = 0. \tag{85}$$

---

**Algorithm 1** FPG-OPS with Uncertainty-Guided Action Blending

---

1: **Input:** Context $\mathcal{C}$, Goal $I_g$, diffusion model $\epsilon_\theta$, task loss $L$, steps $T$, candidates $K$, tail length $M$, guidance scale $\gamma$
2: **Output:** Final blended action $a_{\text{blend}}$
3: **Stage 1: Parallel Diffusion Sampling with FPG-OPS**
4: **for** $k = 1$ **to** $K$ **do**
5:     Initialize $a_T^{(k)} \sim \mathcal{N}(0, I)$      // *Initialize noisy action state*
6:     $U_{\text{score}}^{(k)} \leftarrow 0$      // *Initialize TFDS score*
7:     **for** $t = T$ **down to** $1$ **do**
8:         $\epsilon \leftarrow \epsilon_\theta(\mathcal{C}, a_t^{(k)}, t)$      // *Noise prediction*
9:         Extract OPS basis $W$      // *Residual head $\epsilon \approx W h_\theta(\mathcal{C}, a_t^{(k)}, t)$*
10:        $u_t \leftarrow \nabla_{a_t^{(k)}} L(a_t^{(k)}, \mathcal{C}, t)$      // *Task gradient w.r.t. action state*
11:        $u_h \leftarrow W^\top u_t$      // *Map action-space gradient to OPS coordinates*
12:        Compute $g_h$      // *Latent OPS proxy of Fisher normal direction*
13:        $M_h \leftarrow W^\top W$      // *Pullback metric induced by $W$*
14:        $u_h^\parallel \leftarrow \frac{g_h^\top M_h u_h}{g_h^\top M_h g_h} g_h$      // *Fisher-aligned component*
15:        $u_h^\perp \leftarrow u_h - u_h^\parallel$      // *Fisher-orthogonal component*
16:        $\Delta_t \leftarrow W u_h^\perp$      // *Map projected update back to action space*
17:        $a_{t-1}^{(k)} \leftarrow \text{Solver}(a_t^{(k)}, \epsilon) - \gamma \Delta_t$      // *Guided reverse update*
18:        **if** $t \leq M$ **then**
19:            $U_{\text{score}}^{(k)} \leftarrow U_{\text{score}}^{(k)} + \|\nabla_\mathcal{C} \epsilon_\theta(\mathcal{C}, a_t^{(k)}, t)\|_F^2$      // *TFDS accumulation*
20:        **end if**
21:     **end for**
22:     $\widehat{\mathcal{U}}_k \leftarrow U_{\text{score}}^{(k)}$
23: **end for**
24: **Stage 2: Uncertainty-Guided Action Blending**
25: Cluster $\{a_0^{(k)}\}_{k=1}^K$ using DBSCAN      // *Obtain cluster labels $\{c_k\}$*
26: **for** $k = 1$ **to** $K$ **do**
27:     $w_{\text{local}} \leftarrow \exp(-\eta \widehat{\mathcal{U}}_k)$      // *FDS-based stability weight*
28:     $w_{\text{group}} \leftarrow \frac{|\{j:c_j=c_k\}|}{K}$      // *Cluster typicality weight*
29:     $w_k \leftarrow w_{\text{local}} \cdot w_{\text{group}}$
30: **end for**
31: $a_{\text{blend}} \leftarrow \frac{\sum_{k=1}^K w_k a_0^{(k)}}{\sum_{k=1}^K w_k}$
32: **return** $a_{\text{blend}}$

---

## H.2. Effectiveness with External Guidance (The Constrained Optimization View)

When an external task loss $L(a_t, \mathcal{C}, t)$ (e.g., collision cost or path efficiency) is present, the naive update follows the negative gradient $-\nabla_{a_t} L(a_t, \mathcal{C}, t)$. We show that Fisher-Preserving Guidance (FPG) gives the first-order constrained descent direction for minimizing this loss while adhering to the "Safe Manifold" defined by the Fisher isosurface.

**Theorem H.1** (Optimality of FPG). *Consider the optimization problem of finding a direction $\Delta_t$ that minimizes the task loss $L(a_t + \Delta_t, \mathcal{C}, t)$ locally, subject to the constraint that the Fisher Information remains invariant to the first order:*

$$\min_{\Delta_t} \nabla_{a_t} L(a_t, \mathcal{C}, t)^\top \Delta_t \quad s.t. \quad \nabla_{a_t} \mathcal{I}_t(a_t; \mathcal{C})^\top \Delta_t = 0, \quad \|\Delta_t\| \leq \gamma. \tag{86}$$

**Proof.** We construct the Lagrangian $\mathcal{L}(\Delta_t, \lambda)$ for the optimization direction, ignoring the norm constraint for the derivation of the direction vector first:

$$\mathcal{L}(\Delta_t, \lambda) = \nabla_{a_t} L(a_t, \mathcal{C}, t)^\top \Delta_t + \lambda \big(g_t(a_t)^\top \Delta_t\big). \tag{87}$$

Taking the derivative with respect to $\Delta_t$ and setting it to zero to find the stationary point:

$$\nabla_{\Delta_t} \mathcal{L} = \nabla_{a_t} L(a_t, \mathcal{C}, t) + \lambda g_t(a_t) = 0 \implies \Delta_t^* \propto -\big(\nabla_{a_t} L(a_t, \mathcal{C}, t) + \lambda g_t(a_t)\big). \tag{88}$$

Substituting $\Delta_t^*$ into the constraint equation $g_t(a_t)^\top \Delta_t^* = 0$:

$$g_t(a_t)^\top \left[ -\left( \nabla_{a_t} L(a_t, \mathcal{C}, t) + \lambda g_t(a_t) \right) \right] = 0. \tag{89}$$

$$g_t(a_t)^\top \nabla_{a_t} L(a_t, \mathcal{C}, t) + \lambda \|g_t(a_t)\|^2 = 0. \tag{90}$$

Solving for the Lagrange multiplier $\lambda$:

$$\lambda = -\frac{g_t(a_t)^\top \nabla_{a_t} L(a_t, \mathcal{C}, t)}{\|g_t(a_t)\|^2}. \tag{91}$$

Substituting $\lambda$ back into the expression for $\Delta_t^*$:

$$\Delta_t^* \propto - \left( \nabla_{a_t} L(a_t, \mathcal{C}, t) - \frac{g_t(a_t)^\top \nabla_{a_t} L(a_t, \mathcal{C}, t)}{\|g_t(a_t)\|^2} g_t(a_t) \right). \tag{92}$$

Let $\Delta_{\mathrm{FPG}}$ be the final FPG update step with step size $\gamma$. This confirms that $\Delta_{\mathrm{FPG}}$ is the orthogonal projection of the action-space guidance gradient:

$$\Delta_{\mathrm{FPG}} = -\gamma \cdot \mathrm{Proj}_{\perp g_t} \left( \nabla_{a_t} L(a_t, \mathcal{C}, t) \right). \tag{93}$$

**Conclusion:** This proves that FPG is the first-order optimal descent direction within the tangent space of the Fisher isosurface. It ensures that:

1. **Task Efficiency:** The loss $L$ is reduced by $-\gamma \|\nabla_{a_t} L_\perp\|^2$, where $\nabla_{a_t} L_\perp$ denotes the Fisher-orthogonal component of the action-space task gradient.

2. **Safety Guarantee:** The deviation from the manifold sensitivity is bounded by second-order terms, i.e., $\mathcal{I}_t(a_t + \Delta_{\mathrm{FPG}}; \mathcal{C}) - \mathcal{I}_t(a_t; \mathcal{C}) = O(\gamma^2)$, preventing first-order drift into high-uncertainty or atypical regions often caused by unconstrained guidance.

### H.3. Intrinsic Safety of Fisher-Orthogonal Decomposition (The No-Guidance Case)

Even in the absence of an explicit task loss $L$, decomposing any inherent perturbation (e.g., approximation error, discretization noise) into Fisher-aligned and Fisher-orthogonal components reveals why the orthogonal direction is intrinsically "safe."

Let $\Delta_t$ be an arbitrary perturbation vector applied to the state $a_t$ during the denoising process. We decompose $\Delta_t$ into two orthogonal components:

$$\Delta_t = \Delta_\| + \Delta_\perp, \quad \text{where } \Delta_\| \parallel g_t(a_t), \quad \Delta_\perp \perp g_t(a_t). \tag{94}$$

**Proposition H.2** (First-Order Sensitivity Stability). *A perturbation $\Delta_\|$ along the Fisher-normal direction tends to induce a larger first-order change in the local sensitivity field, whereas $\Delta_\perp$ preserves this sensitivity to first order.*

**Proof.** Consider the predictive distribution $p_\theta(a_0 | a_t, \mathcal{C})$ parameterized by the diffusion backbone. The local change in this distribution caused by a perturbation $\Delta_t$ can be measured by the Kullback-Leibler (KL) divergence, approximated by the quadratic form of the Fisher Information Matrix (FIM) $\mathbf{F}_{a_t}$:

$$D_{KL}\left( p_\theta(\cdot | a_t, \mathcal{C}) \,\|\, p_\theta(\cdot | a_t + \Delta_t, \mathcal{C}) \right) \approx \frac{1}{2} \Delta_t^\top \mathbf{F}_{a_t} \Delta_t. \tag{95}$$

In our context, the scalar FDS $\mathcal{I}_t(a_t; \mathcal{C})$ acts as a practical proxy for local sensitivity. The gradient $g_t(a_t) = \nabla_{a_t} \mathcal{I}_t(a_t; \mathcal{C})$ points in the direction where the sensitivity changes most rapidly.

**The Danger of $\Delta_\|$ (Normal Component):** Moving along $g_t(a_t)$ implies moving from a region of regular sensitivity to a region of different sensitivity, often higher and less stable:

$$\mathcal{I}_t(a_t + \Delta_\|; \mathcal{C}) \approx \mathcal{I}_t(a_t; \mathcal{C}) + \|\nabla_{a_t} \mathcal{I}_t(a_t; \mathcal{C})\| \cdot \|\Delta_\|\|. \tag{96}$$

A rapid increase in Fisher sensitivity implies a rapid increase in the local Lipschitz behavior of the score function. This can lead to numerical instability in the reverse diffusion solver, causing "Manifold Explosion."

**The Safety of $\Delta_\perp$ (Tangential Component):** By definition, $g_t(a_t)^\top \Delta_\perp = 0$. The change in sensitivity is:

$$\mathcal{I}_t(a_t + \Delta_\perp; \mathcal{C}) \approx \mathcal{I}_t(a_t; \mathcal{C}) + g_t(a_t)^\top \Delta_\perp + O(\|\Delta_\perp\|^2) = \mathcal{I}_t(a_t; \mathcal{C}) + O(\|\Delta_\perp\|^2). \tag{97}$$

Because the sensitivity remains locally constant to first order, the conditioning of the score function $\epsilon_\theta(\mathcal{C}, a_t, t)$ remains stable. The perturbation $\Delta_\perp$ represents a movement **along the data manifold** (changing semantic content, e.g., moving forward vs. turning) rather than **off the manifold** (changing generation quality or reliability).

**Conclusion:** In the absence of external guidance, if we must process a perturbation or weak prior, projecting it onto the Fisher-orthogonal subspace ($\Delta_\perp$) acts as a **Stabilizing Filter**. It preserves the **Typicality** of the sample by keeping the trajectory within a local "Trust Region" of stable Fisher sensitivity, preventing transitions into high-sensitivity and unreliable states.

## I. Pseudocode for FPG-OPS with Action Blending

The pseudocode Algorithm 1 summarizes the Fisher-preserving guidance and uncertainty-aware action blending procedure for diffusion policy inference:

