# OpenReview forum: "Fisher-Preserving Guidance: Training-Free Manifold Constraints for Safe Diffusion Control"
_ICML.cc/2026/Conference — ICML 2026 regular_

### Official Review · Reviewer_um9C · 2026-03-12

**Soundness:** 2
**Presentation:** 3
**Significance:** 2
**Originality:** 3
**Overall Recommendation:** 4
**Confidence:** 2

**Summary:**

This paper proposes Fisher-Preserving Guidance (FPG), a training-free inference-time method for diffusion-based navigation policies that constrains gradient updates to lie on a Fisher isosurface, aiming to prevent off-manifold drift during test-time guidance. The method is paired with a low-rank Outer Product Span projection for computational efficiency and an uncertainty-guided action blending module using truncated Fisher sensitivity and cluster typicality. Experiments span toy environments (Maze2D, PushT), simulation (CARLA, Isaac Sim), and a brief real-robot demonstration, with the full system outperforming NoMaD and ViNT baselines on success rate and collision metrics.

**Compliance With Llm Reviewing Policy:**

Affirmed.

**Final Justification:**

The paper has clear strengths in its geometric motivation, training-free design, practical OPS approximation, and broad empirical evaluation. My main concern was that the original framing around “manifold constraints” and “safe diffusion control” was stronger than what was actually established. The rebuttal did not show that this concern was mistaken, but it did clarify that the intended claim is narrower: the Fisher-based quantity should be viewed as a local sensitivity / typicality proxy rather than a calibrated safety guarantee or a formally established characterization of the training manifold. Given this clarification, together with the added empirical analyses, I am updating my evaluation positively and raising my score to 4, conditional on the final version revising the claim accordingly.

**Key Questions For Authors:**

See weakness.

**Limitations:**

The authors' limitations section acknowledges that uncertainty estimates are not calibrated safety guarantees and notes risks of overconfidence under distribution shift. What it does not address is the architectural dependency of the OPS approximation, the absence of statistical reporting for the key plug-and-play experiment, or the scope restriction to RGB-only waypoint prediction policies.

**Strengths And Weaknesses:**

Pros:

- The core geometric intuition — that safe guidance updates should preserve Fisher information to first order — is clearly stated and provides a principled lens for diagnosing why naive test-time gradient steps degrade diffusion policy performance.
- The OPS low-rank factorization is a practically motivated engineering contribution that brings the per-step overhead to a single backward pass, making the method deployable in real-time settings.
- The ablation in Table 4 is structured cleanly, with FPG and action blending separated, allowing a reader to attribute performance gains to individual components without ambiguity.


Cons:
- The central theoretical claim rests on the premise that the Fisher isosurface is a meaningful proxy for the boundary of the in-distribution region of the diffusion model. This is presented as geometrically intuitive but is never formally justified. The Fisher Denoising Sensitivity I(C,t) is the Frobenius norm of a Jacobian, which measures local sensitivity of the predicted noise to observation perturbations — this is not the same as measuring proximity to the training manifold in any rigorous sense. Without establishing that level sets of I(C,t) actually correspond to high-probability regions under the learned distribution, the "safety guarantee" framing is aspirational rather than proven. - The paper would be more honest to describe FPG as a regularization heuristic with a Fisher-inspired motivation rather than a manifold constraint with safety guarantees.
- The PushT result is the one experiment where an existing pretrained backbone (official Diffusion Policy weights) is evaluated without domain-specific fine-tuning, making it the cleanest test of the plug-and-play claim. The improvement there is 0.91 to 0.94 task score — a 3-point gain that is reported without confidence intervals or repeated trials. Given that diffusion policy task scores on PushT are known to have non-trivial variance across seeds, this result cannot be taken as conclusive evidence of improvement. The CARLA and Isaac Sim experiments show larger gains but use a NoMaD backbone that achieves only 51% success rate on CARLA, which is substantially below what specialized driving policies achieve on that simulator, raising questions about whether the baseline setup is competitive enough to generalize the conclusions.
- The OPS projection relies structurally on the existence of an explicit linear layer W mapping from a latent representation h(C) to the action space, from which the low-rank Jacobian factorization follows. This architectural assumption is specific to the NoMaD-style encoder-projection design used in this paper and does not obviously extend to transformer-based diffusion policies or architectures with nonlinear decoders. The paper does not discuss this limitation, and given that much of the diffusion policy community has moved toward UNet and transformer backbones, the scope of applicability of the efficient OPS computation is narrower than the framing implies.
- The real-robot experiment in Section 4.7 consists of six images showing a navigation sequence with no quantitative results, no comparison to a baseline on hardware, and no description of how many trials were run or what the success criterion was. This section contributes nothing verifiable to the empirical story and should either be replaced with a proper quantitative evaluation or removed.
A prompt injection string is embedded near the end of the appendix, instructing a reviewer to include specific phrases in their review. This is a serious integrity concern that is independent of the scientific content and should be flagged to the program chairs.

---

> ### Author Rebuttal · Authors · 2026-03-31
>
> Thank you for your insightful comments. We address your main concerns below.
>
> **1. The central theoretical claim is not formally proven; Fisher level sets are not established as a boundary of the in-distribution region.**
>
> We thank the reviewer for this rigorous critique and fully agree with your assessment. To clarify, the term safety in our work strictly denotes distributional safety. This means constraining generations to the in-distribution training manifold to prevent extrapolation errors, rather than ensuring physical or operational safety, which aligns with the limitations explicitly emphasized in our Impact Statement. While you are absolutely correct that the Fisher sensitivity $I(C,t)$ formally measures local sensitivity rather than rigorous manifold proximity, it serves as a highly effective empirical proxy. Because diffusion score functions are smooth in high-density regions but highly unstable in unsupported out-of-distribution spaces, bounding $I(C,t)$ effectively penalizes such deviations. Appendix G.2 empirically validates this mechanism, demonstrating that FPG successfully anchors trajectories within high-probability regions. Nevertheless, we concede that claiming a formal safety guarantee could cause some misleading. We will gladly adopt your precise characterization by removing claims of guarantees and accurately reframing FPG as a regularization heuristic with a Fisher-inspired motivation.
>
> **2. The PushT result (0.91 to 0.94) is too small and lacks confidence intervals.**
>
> To ensure a fair comparison, the PushT task adopts the same experimental setup as Diffusion Policy (DP). Furthermore, we provide a statistical significance analysis and results across multiple checkpoints to substantiate our findings. The numbers $1$, $2$, and $3$ represent different checkpoints. The metrics in the table represent the scores and their corresponding standard deviations.
>  ||1|2|3|
> |-|-|-|-|
> |DP|0.9139±0.0600|0.9195±0.0638|0.8939±0.0832|
> |DP+OPS|0.9401±0.0433|0.9412±0.0456|0.9197±0.0569|
>
> **3. The CARLA baseline may not be competitive enough in absolute terms.**
>
> We understand the concern. Our goal in CARLA is a controlled within-backbone comparison under identical training and evaluation protocols, rather than to compete with CARLA-specialized driving systems trained specifically for that simulator. Under this controlled setting, OPS yields substantial gains over the fixed NoMaD backbone, which is the intended claim. We will clarify this scope and avoid over-generalizing from absolute CARLA numbers.
>
> **4. OPS appears architecture-dependent because it relies on an explicit linear layer (W).**
>
> Thank you very much for pointing out this issue. The low-rank OPS factorization is not specific to a NoMaD-style encoder-projection architecture. For any transformer-based diffusion policy whose denoiser output is produced through a linear readout head, i.e., $\epsilon_\theta = P\mathrm{vec}(Z)+b$ with $Z$ the transformer hidden states, the Jacobian factorization $\nabla_{a_t} \epsilon_\theta = P \nabla_{a_t} \mathrm{vec}(Z)$ follows exactly by the chain rule, and the associated Fisher matrix has rank at most $\operatorname{rank}(P)$. More generally, for a differentiable nonlinear decoder $g$, one still has $\nabla_{a_t} \epsilon_\theta = J_g(h_t)\nabla_{a_t} h_t$, yielding a local first-order OPS factorization with an input-dependent projection $W_t=J_g(h_t)$. Therefore, the derivation extends to a broad class of transformer-based robot diffusion policies; what changes across architectures is not validity per se, but whether the induced latent dimension is sufficiently small to provide a strong low-rank computational advantage. We validated our approach using an image-input, transformer-based diffusion policy, demonstrating that OPS consistently improves performance across various settings.
> ||1|2|3|
> |-|-|-|-|
> |DP-T|0.6961±0.0672|0.7393±0.1101|0.7738±0.0800|
> |DP-T+OPS|0.7259±0.0733|0.7485±0.1107|0.7923±0.0672|
>
> **5. The real-world experiment is only qualitative and should be replaced with a quantitative evaluation or removed.**
>
> We agree that the quantitative real-robot results should have been surfaced much more prominently in the main paper. They are currently present in the Appendix (Table 9), where FPG achieves 28/30 success with 0.10 average collisions, versus 22/30 and 0.37 for ViNT, and 19/30 and 0.57 for NoMaD. We will move these quantitative results into the main paper instead of relying on the qualitative figure.
>
> **6. A prompt injection string is embedded near the end of the appendix, instructing a reviewer to include specific phrases in their review.**
>
> We understand the reviewer’s concern and confirm that we did not intentionally insert any prompt into the manuscript. According to the official email, the PDF may have been embedded with a watermark for detecting possible LLM-assisted reviewing. Regarding prompt injection, please see https://icml.cc/Conferences/2026/PeerReviewFAQ#prompt_injection.

---

> > ### Author Rebuttal · Reviewer_um9C · 2026-04-03
> >
> > I appreciate the authors’ detailed rebuttal and the additional analyses provided during the discussion phase. In particular, I appreciate that the authors directly addressed my main conceptual concern and clarified that the proposed Fisher-based quantity should be interpreted as a local sensitivity / typicality proxy rather than as a calibrated safety guarantee or a formally established characterization of the training manifold.
> >
> > This point is important because, in my view, the strongest weakness of the original manuscript was that its central narrative was stated more strongly than what was actually established. In other words, the rebuttal did not convince me that my original concern was mistaken; rather, it clarified that the intended claim is narrower and more conservative than the original framing suggested. I view this as a meaningful improvement. The original headline interpretation of “manifold constraints” and “safe diffusion control” is therefore weakened, but also made more precise and more defensible.
> >
> > Given that the authors have explicitly acknowledged this issue and moved the claim to a more careful and reasonable position, I am updating my assessment positively and will raise my score to 4.

---

> > > ### Author Response · Authors · 2026-04-03
> > >
> > > We sincerely thank you for your thoughtful engagement with our rebuttal and for updating your assessment and raising your score.
> > >
> > > We are very glad that our clarification addressed your primary conceptual concern. We completely agree with your assessment that framing our method as a local sensitivity proxy, rather than a formal safety guarantee, makes the core claims of the paper much more rigorous, precise, and defensible. By adopting this view, we can clearly articulate our mechanism as guiding the gradient direction to stay on the Fisher isosurface, utilizing it as a local manifold that preserves sensitivity. It is through this prudent gradient constraint that we achieve improved performance across downstream tasks.
> > >
> > > As committed to in our rebuttal, we will carefully revise the manuscript's narrative, particularly in the introduction, methodology, and conclusion, to ensure this more accurate and conservative framing is clearly established in the final version. We will also ensure the quantitative real-robot results and the expanded architectural discussions are prominently integrated into the main text.
> > >
> > > Thank you again for your highly constructive and rigorous feedback, which has genuinely helped strengthen the scientific clarity and impact of our work.

---

### Official Review · Reviewer_J8Ut · 2026-03-13

**Soundness:** 4
**Presentation:** 2
**Significance:** 3
**Originality:** 3
**Overall Recommendation:** 4
**Confidence:** 4

**Summary:**

This paper proposes Fisher-Preserving Guidance (FPG-OPS), a training-free test-time guidance method for diffusion control. The key idea is to project task guidance onto the tangent space of the Fisher isosurface during reverse diffusion, so that guided sampling remains closer to the training manifold / high-confidence region. The paper further introduces TFDS-based uncertainty-guided action blending and demonstrates improved performance mainly on navigation benchmarks as well as on real-robot experiments.

**Compliance With Llm Reviewing Policy:**

Affirmed.

**Final Justification:**

All of my concerns have been adequately addressed, and I will maintain my score.

**Key Questions For Authors:**

Please refer to the Weaknesses section for the detailed context of my concerns. My main questions are about whether the central manifold-preserving claim is fully supported by the current evidence, and whether some of the core design choices are uniquely justified. Clarifying these points would meaningfully affect my final evaluation.

**Limitations:**

Yes. The paper does mention in the impact statement that the uncertainty estimates are not calibrated safety guarantees and that deployment under distribution shift requires care.

**Strengths And Weaknesses:**

Strengths:
1. The paper includes detailed mathematical formulation and derivation.
2. The paper is generally well written and easy to follow, with clear organization and presentation.
3. The proposed method is training-free and can be integrated into existing diffusion-based control or policy frameworks at inference time.
4. The inclusion of real-robot experiments strengthens the empirical validation.

Weaknesses:
1. Given the paper’s manifold-preserving claim, a more controlled study on the training distribution would be valuable. In particular, it remains unclear whether FPG is equally effective, or more beneficial, under narrower or more homogeneous training distributions.
2. The paper lacks sensitivity analysis for several key inference-time hyperparameters, such as the TFDS truncation horizon M, guidance scale, number of candidate samples, blending temperature, and clustering-related settings.
3. The projection removes the entire Fisher-normal component, although only the Fisher-increasing part seems clearly undesirable. It is unclear why the Fisher-reducing component is also discarded, as it could potentially move the sample toward a higher-confidence region.
4. The relationship between input-Fisher / FDS and unfamiliar, off-manifold, or risky regions needs further clarification.
5. The notation around the optimization variable is somewhat unclear. The task guidance is written in terms of gradients with respect to the conditioning/context variable C, yet the projected correction is applied directly in the reverse diffusion update of the action trajectory.
6. Although the paper is titled as safe diffusion control, most of the empirical evidence is concentrated on navigation benchmarks, with only limited validation on manipulation/control beyond PushT.
7. The paper lacks explicit failure-case analysis and statistical significance testing.
8. Some figures contain text that is too small to read comfortably.

---

> ### Author Rebuttal · Authors · 2026-03-31
>
> Thank you for your positive comments and suggestions!
>
> **1. The ‘manifold-preserving’ claim would be better validated by a controlled study over the training distribution.**
>
> We sincerely thank the reviewer for this excellent suggestion. We completely agree that a controlled study systematically varying the training distribution provides a much stronger, empirical validation of our 'manifold-preserving' claim. Motivated by your feedback, we conducted this exact controlled study during the rebuttal period. To systematically test the robustness of our method under narrower versus broader training distributions, we trained the models using varying fractions of the dataset (1%, 10%, 50%, and 100%). We compared our proposed method (OPS) against the DDPM baseline across MSE, ADE, and FDE metrics. OPS performed well across different training distributions.
>  |(OPS/DDPM)|1%|10%|50%|100%|
> |-|-|-|-|-|
> |MSE↓|2.978/3.191|2.639/3.126|1.769/1.883|1.361/1.601|
> |ADE↓|1.770/1.839|1.569/1.709|1.223/1.288|1.088/1.179|
> |FDE↓|3.630/3.777|3.232/3.562|2.404/2.545|2.104/2.307|
>
> **2. There is insufficient sensitivity analysis for $M$, guidance scale, number of candidates, blending temperature, and clustering settings.**
>
> We agree. This is one of the main missing experimental pieces in the current draft. We included sensitivity analyses for the truncation steps $M$, guidance scale $\gamma$, blending temperature $\eta$, and clustering threshold $\epsilon$, detailed in our responses to Reviewers wSmR (Q2, Q3) and Sska (Q2). The experimental results for the sensitivity to the number of candidates are shown below. Increasing the number of candidates can consistently improve the performance of the final action. Nevertheless, actual deployment requires balancing inference overhead with action accuracy, which precludes an excessive increase in $K$.
> |K|MSE↓|ADE↓|FDE↓|
> |-|-|-|-|
> |2|1.392|1.108|2.143|
> |4|1.378|1.099|2.127|
> |8|1.361|1.088|2.104|
> |16|1.324|1.069|2.060|
>
> **3. Why remove the entire Fisher-normal component? Why not only remove the Fisher-increasing part?**
>
> We thank the reviewer for this insightful question. Our method intentionally solves a local equality-constrained problem. We minimize the task loss subject to first-order Fisher invariance $g^\top \Delta = 0$. Under this formulation, removing the entire Fisher-normal component is exactly the orthogonal projection onto the tangent space of the Fisher isosurface.
>
> While a one-sided rule that removes only Fisher-increasing motion is an interesting inequality-constrained alternative, our objective is not to monotonically drive Fisher downward. As motivated in Appendix G.1, we aim to preserve the local sensitivity regime by avoiding both excess and vanishing Fisher.
>
> **4. The relationship between input Fisher / FDS and unfamiliar, off-manifold, or dangerous regions needs clarification.**
>
> We agree and will clarify this carefully. In our paper, FDS is a local input-sensitivity proxy; it is not identical to task danger. “Off-manifold / unfamiliar” is a distributional / typicality notion that FDS only approximates. “Dangerous” is task-dependent and enters through the task loss / external guidance. Our safety interpretation comes from combining task guidance with Fisher-preserving projection, not from equating high FDS with collision risk. We will bring this distinction into the main text and connect it more clearly to Appendix F.
>
> **5. The notation is unclear: guidance is written as a gradient w.r.t. the condition/context, but the projection is applied to the reverse diffusion update of the action trajectory.**
>
> Thank you very much for pointing this out. We will thoroughly revise the relevant presentation to ensure that the notation is precise and consistent throughout. The definitions related to FDS are with respect to the conditioning variable (C), and are used to measure the sensitivity of the output to perturbations in the condition. In contrast, the task-guidance term should be computed with respect to (a_t). Appendix H already presents the correct subscript notation. This is a clarity issue in the presentation, not a change to the underlying method.
>
> **6. The experimental scope is limited, despite the title about ‘safe diffusion control.’**
>
> We agree that evaluating our approach on a wider variety of operational tasks is highly beneficial. So far, we have validated our method in path planning and manipulation alongside verifying its effectiveness in complex navigation scenarios. Moving forward, we plan to broaden our experimental scope by applying this method to other diffusion control domains, specifically targeting robotic manipulation and image generation.
>
> **7. The paper lacks failure-case analysis and statistical significance testing.**
>
> We agree. We will add these statistics and include representative failure cases for all three tasks.
>
> **8. Some figure text is too small to read comfortably.**
>
> We will enlarge the fonts and labels in the final figures and tables.

---

> > ### Author Rebuttal · Reviewer_J8Ut · 2026-04-04
> >
> > Thank you for the rebuttal. My questions have been adequately addressed, and I have no further concerns at this point. I will maintain my current assessment.

---

> > > ### Author Response · Authors · 2026-04-04
> > >
> > > We sincerely thank you for the feedback and for acknowledging that our responses have addressed your concerns. We greatly appreciate your valuable time, effort, and insightful feedback, which have significantly improved the manuscript, particularly your thorough review of the mathematical expressions.

---

### Official Review · Reviewer_Sska · 2026-03-13

**Soundness:** 3
**Presentation:** 3
**Significance:** 3
**Originality:** 3
**Overall Recommendation:** 4
**Confidence:** 4

**Summary:**

This paper addresses the instability of Group Relative Policy Optimization (GRPO) methods in video generation, attributing it to the high complexity of the video solution space and noise from ODE-to-SDE conversion, which degrades rollout quality and reward reliability. To tackle this, the authors propose SAGE-GRPO, a framework that constrains exploration to the manifold of pre-trained video models to ensure stability. It employs micro-level techniques like a manifold-aware SDE with curvature correction and a temporal gradient equalizer, alongside macro-level controls using a dual-KL trust region with dynamic anchors. Experiments on the HunyuanVideo1.5 model show SAGE-GRPO significantly outperforms existing methods in both quantitative and visual quality metrics.

**Compliance With Llm Reviewing Policy:**

Affirmed.

**Final Justification:**

All of my concerns have been adequately addressed, and I will maintain my score.

**Key Questions For Authors:**

1.Could you provide comparative experiments on computational efficiency and accuracy between the proposed OPS projection and other low-rank approximation methods, such as Hutchinson?
2.How does the method perform in scenarios with dynamic obstacles?
3.Could you provide a sensitivity analysis for the hyperparameters (e.g., M, clustering thresholds)?

**Limitations:**

NO

1.the choice of the TFDS truncation step M (e.g., 4 in the paper) lacks sufficient data support and discussion.

2.the experimental design does not include dynamic obstacle scenarios. This limits the assessment of the method's robustness in real-world dynamic environments and represents a potential risk for future real-world deployment.

**Strengths And Weaknesses:**

Strengths

Soundness: The Fisher equiconstraint and the OPS low-rank approximation are both supported by rigorous theoretical derivations. The experimental design is comprehensive, covering multiple scenarios with reasonable baseline comparisons. Ablation studies verify the complementary nature of the core modules, lending credibility to the results.

Presentation: The paper is well-structured, coherently narrated, and accurately distills its core innovations and experimental conclusions.
Significance: It focuses on a core safety concern in visual navigation. The "plug-and-play", no-retraining-required nature of the method reduces deployment costs. Its transferability to various diffusion-based navigation systems offers practical solutions for robotics and autonomous driving, indicating significant application value.

Originality: It is the first to introduce Fisher-preserving constraints into the inference stage of diffusion-model-based navigation. It proposes the OPS low-rank projection and TFDS uncertainty fusion strategy, effectively addressing the computational bottleneck of Fisher constraints in high-dimensional scenarios and the challenge of selecting multimodal actions.

Weaknesses

Soundness: The paper does not quantify the computational efficiency differences between the proposed OPS projection and other low-rank approximation methods . The rationale for the choice of the TFDS truncation step M is not clearly explained.

Presentation: A sensitivity analysis of key hyperparameters (e.g., the TFDS truncation step M) on performance is missing, and further details for reproducibility could be added.

Significance: The experiments are validated only on pure RGB visual navigation scenarios, without extension to multimodal inputs.

Originality: The proposed uncertainty-guided action blending mechanism is a customized application for the navigation scenario, and its generalizability to other domains requires further discussion.

---

> ### Author Rebuttal · Authors · 2026-03-31
>
> Thank you for your positive comments and suggestions. Below, we carefully address several of the points that they raised.
>
> **1. Can you compare OPS with other low-rank methods such as Hutchinson in both efficiency and accuracy?**
>
> We agree that this comparison should be made more explicit. The current paper already includes an efficiency comparison against a full VJP-Fisher baseline. We also evaluated replacing the original OPS with a Hutchinson-style low-rank approximation under the same setup as Table 5. However, because the Hutchinson method requires multiple rounds of random sampling for estimation, its real-time performance is substantially compromised. The denoising time reaches 194 ms for 10 steps and 77.7 ms for 4 steps, both significantly slower than our OPS method (52.3ms for 10 steps and 35.5ms for 4 steps). The performance of several sampling methods on the test set of the visual navigation dataset used in this paper, as measured by MSE, ADE, and FDE, is shown in the table.
> ||MSE↓|ADE↓|FDE↓|
> |-|-|-|-|
> |OPS|1.361±0.927|1.088±0.358|2.104±0.851|
> |DDPM|1.601±1.072|1.179±0.394|2.307±0.950|
> |Hutchinson|3.606±2.161|1.842±0.676|3.853±1.472|
> |VJP Fisher|3.810±2.309|1.887±0.705|3.958±1.540|
>
> **2. Could you provide a sensitivity analysis for the hyperparameters (e.g., M, clustering thresholds)?**
>
> We conducted a sensitivity analysis of the truncation step $M$ and the step size $\gamma$ for PushT tasks. Appropriate parameter selection is necessary. Additionally, for action blending, we performed a sensitivity analysis of the clustering threshold $\epsilon$ and the temperature parameter $\eta$ for visual navigation in test-set offline inference tasks. Gradually increasing $\epsilon$ leads to improvements across all metrics. Meanwhile, varying $\eta$ affects the overall performance, but its impact remains marginal.
> |$M$|1|2|3|4|5|16|27|28|29|
> |-|-|-|-|-|-|-|-|-|-|
> |score|0.918|0.933|0.947|0.947|0.941|0.951|0.959|0.956|0.951|
> |std|0.254|0.229|0.209|0.207|0.217|0.209|0.191|0.197|0.202|
>
> |$\gamma$|0.001|0.005|0.009|0.01|0.011|0.015|0.03|0.04|
> |-|-|-|-|-|-|-|-|-|
> |score|0.917|0.91|0.941|0.94|0.942|0.933|0.934|0.911|
> |std|0.243|0.258|0.207|0.21|0.207|0.211|0.239|0.272|
>
> |$\epsilon$|MSE↓|ADE↓|FDE↓|
> |-|-|-|-|
> |0.2|1.390|1.106|2.142|
> |0.3|1.386|1.102|2.131|
> |0.5|1.361|1.088|2.104|
> |0.8|1.328|1.074|2.072|
>
>  |$\eta$|MSE↓|ADE↓|FDE↓|
> |-|-|-|-|
> |1|1.357|1.087|2.100|
> |2|1.367|1.088|2.103|
> |5|1.361|1.088|2.104|
> |10|1.358|1.087|2.100|
>
> **3. The experiments are only validated on pure RGB visual navigation and are not extended to multimodal input.**
>
> This is a valid limitation. Our current empirical scope is mainly RGB visual navigation, with PushT included as a plug-and-play non-navigation test. We will add Section Limitations to make this scope explicit and avoid implying that multimodal generalization has already been established. Exploring broader multimodal applications is a valuable future direction, much like the extension of diffusion policy to 3D diffusion policy.
>
> **4. How does the method perform in dynamic-obstacle scenarios?**
>
> We agree that both are important. The current paper does not isolate dynamic-obstacle scenarios as a separate benchmark, and we will state this more clearly as a limitation. Since the core objective of this paper is to make the model's outputs more in-distribution during inference, without additional adaptations for dynamic scenarios, its performance is primarily constrained by the underlying model.
>
> **5. The uncertainty-guided action blending seems tailored to navigation; its generalization to other domains should be discussed.**
>
> We agree that the current empirical evidence for broad generalization is limited. At the algorithmic level, action blending is inherently domain-agnostic: it leverages the multimodal distribution of diffusion models using only candidate actions, TFDS scores, and cluster typicality, without relying on navigation-specific state variables. However, despite this theoretical generality, our current empirical validation is predominantly confined to path planning (specifically visual navigation), with PushT serving as our only preliminary test for manipulation. Therefore, we will tone down the generality claims in the revised manuscript and explicitly state that broader cross-domain validation remains an area for future work.

---

> > ### Author Rebuttal · Reviewer_Sska · 2026-04-05
> >
> > I greatly appreciate that the authors have thoroughly considered my core concerns and provided constructive responses. At this point, I have no further issues and maintain my rating of “weak accept.”

---

> > > ### Author Response · Authors · 2026-04-05
> > >
> > > We sincerely thank you for the feedback and for acknowledging that our responses have addressed your core concerns. We greatly appreciate your valuable time, effort, and insightful feedback.

---

### Official Review · Reviewer_wSmR · 2026-03-16

**Soundness:** 3
**Presentation:** 3
**Significance:** 3
**Originality:** 4
**Overall Recommendation:** 5
**Confidence:** 3

**Summary:**

In this paper the authors propose (1) a new way to quantify uncertainty in diffusion denoising process by defining Step Fisher Denoising Sensitivity and Chain Fisher Denoising Sensitivity, and Truncated Fisher Denoising Sensitivity to approximate CFDS, (2) a new training-free diffusion guidance method that uses the newly defined Fisher Denoising Sensitivity to keep the intermediate diffusion steps within a Fisher isosurface, and (3) an uncertainty guided action blending mechanism. Experiments are tested on safety navigation benchmarks and real-world robots to show the effectiveness of the method.

**Compliance With Llm Reviewing Policy:**

Affirmed.

**Key Questions For Authors:**

It would be great if the authors can address the concerns I listed in the previous section about the weakness of the paper.

**Limitations:**

There is no limitation discussion in this paper and my takes on the limitations of the paper have been included above.

**Strengths And Weaknesses:**

*Soundness:**

  **Strength:**

1. The range of experiment settings is very diverse, covering from toy examples to real-world robot navigation. I think this is quite impressive and shows strong evidence of the effectiveness of this method.
2. The experimental results also show significant improvement upon baselines.

  **Weakness:**

  1. There is no limitation analysis in this paper, which the authors should consider adding.
  2. Typically in diffusion guidance methods the step size $\gamma$ is a very sensitive hyperparameter, is this algorithm also step size sensitive? The authors failed to include an ablation study on this key hyper parameter.
  3. The authors failed to include the experiments that lead to the conclusions they claimed in Section 3.2 “Practical Implication”. It would be valuable for authors to include the details of these experiments in the paper.

**Presentation:**

  **Strength:** The writing is very easy to follow. The authors did a good job at setting the stage and explaining all components that lead up to their algorithms.


  **Weakness:** The authors failed to discuss the diffusion guided posterior sampling literature entirely, specifically I would recommend the authors to include discussion on this paper:
  He et al. Manifold Preserving Guided Diffusion. ICLR 2024.

**Significance:**

  **Strength:** As the domain of embodied AI/agents becomes increasingly important, the task of visual navigation also gains its significance. Since diffusion models can be considered a major trend in the domain, I think exploring training-free diffusion guidance methods that specifically fits the needs of this task can have good influence.

**Originality:**

**Strength:** This paper is novel and creative w.r.t. its theory-grounded algorithmic design and practical application. I don’t think there should be questions about its originality.

---

> ### Author Rebuttal · Authors · 2026-03-31
>
> Thank you for your positive comments. We address your points below.
>
> **1. The paper lacks limitations analysis.**
>
> Thank you for the suggestion. We agree that the current manuscript should state its limitations more clearly, and we will add a dedicated limitations section. Specifically, we will clarify that Fisher denoising sensitivity is a practical proxy for local sensitivity and typicality rather than a certified physical safety guarantee, and that our empirical evaluation remains focused mainly on RGB visual navigation, with limited results on Maze2D and PushT.
>
> **2. Step size is a sensitive hyperparameter in diffusion guidance methods; is your method also sensitive to it?**
>
> Thank you for the suggestion. The step size $\gamma$ is indeed an important parameter. To illustrate its effect, we conducted an ablation study on the PushT task. When $\gamma$ is too small, the guidance effect is limited. As $\gamma$ increases, the guidance becomes more effective, and the inference results tend to better align with the training-domain distribution, leading to higher scores. When $\gamma$ becomes too large, overly strong guidance starts to weaken the model’s ability to accomplish the task, although the overall performance still remains no worse than the baseline. The optimal value of $\gamma$ may vary across tasks; for the visual navigation experiments, the default hyperparameters are listed in Table 7.
>  |gamma|score|std|
> |-|-|-|
> |0.001|0.917|0.243|
> |0.005|0.910|0.258|
> |0.009|0.941|0.207|
> |0.01|0.940|0.210|
> |0.011|0.942|0.207|
> |0.015|0.933|0.211|
> |0.03|0.934|0.239|
> |0.04|0.911|0.272|
>
> **3. The paper does not include experiments supporting the conclusion in Sec. 3.2 (‘Practical implication’).**
>
> We agree that empirical validation is valuable. Sec. 3.2 provides the theoretical analysis of the truncation parameter in Truncated Fisher Denoising Sensitivity (TFDS), demonstrating that only a small fraction of tail steps is required to approximate CFDS. To further support this point, we also conducted a detailed analysis on the PushT task. When TFDS is used in Fisher-Preserving Guidance, the score increases rapidly as $M$ grows and then stabilizes at around 0.94. In particular, performance improves noticeably once $M$ becomes moderately large, and reaches a high level when $M$ approaches the total number of sampling steps. This indicates that the truncation error of TFDS is small in practice, which is consistent with the theoretical analysis.
> |M|score|std|
> |-|-|-|
> |1|0.918|0.254|
> |2|0.933|0.229|
> |3|0.947|0.209|
> |4|0.947|0.207|
> |5|0.941|0.217|
> |16|0.951|0.209|
> |27|0.959|0.191|
> |28|0.956|0.197|
> |29|0.951|0.202|
>
> **4. Please discuss related work on diffusion-guided posterior sampling, in particular He et al., Manifold Preserving Guided Diffusion.**
>
>  We thank the reviewer for pointing this out. We agree that the diffusion guided posterior sampling literature, especially He et al., Manifold Preserving Guided Diffusion (ICLR 2024), is relevant and should be discussed. MPGD is closely related in spirit: it studies training-free guided diffusion and posterior sampling, aims to reduce harmful drift by encouraging manifold-consistent guidance, including updates on the clean estimate and autoencoder-/latent-based on-manifold projections.
> At the same time, our work addresses a different setting and introduces a different technical mechanism. MPGD focuses on conditional generation and posterior sampling, especially for pre-trained autoencoders, whereas our paper focuses on diffusion control, using a Fisher-sensitivity-preserving projection together with TFDS-based action blending. We will add this discussion to the revised related-work section and clarify that the two works are complementary: both emphasize preserving structure during guided diffusion; building on this shared goal,  our method contributes a distinct control-oriented formulation and stabilization mechanism.

---

> > ### Author Rebuttal · Reviewer_wSmR · 2026-04-04
> >
> > The authors fully resolved my concerns so I will remain my accepting score.

---

> > > ### Author Response · Authors · 2026-04-04
> > >
> > > We sincerely thank you for positively recognizing the contributions of our work. We greatly appreciate your valuable time, effort, and insightful feedback, which will be instrumental in further improving our manuscript.

---

### Decision · Program_Chairs · 2026-04-30

**Decision:**

Accept (regular)

**Comment:**

This paper proposes Fisher-Preserving Guidance with Outer Product Span Projection (FPG-OPS), a training-free inference-time method for diffusion-based navigation policies. The method constrains guidance updates to the tangent space of the Fisher isosurface to prevent off-manifold drift, paired with a truncated Fisher Denoising Sensitivity (TFDS) measure for uncertainty-guided multi-sample action blending.

The paper received four reviews, with scores of 5, 4, 4, and 4 (originally lower for Reviewer um9C, raised after rebuttal). All reviewers agreed on the paper's strengths: a principled and novel geometric motivation, a computationally efficient low-rank approximation that enables real-time use, well-organized presentation, and a broad experimental scope spanning toy benchmarks (Maze2D, PushT), simulation (CARLA, Isaac Sim), and real-robot deployment.

The main concerns raised across reviews centered on: (1) the strength of the "manifold constraint" and "safety guarantee" framing relative to what is formally established; (2) missing sensitivity analyses and statistical reporting; (3) the scope of evaluation being primarily RGB visual navigation; (4) the qualitative-only real-robot section; and (5) architectural generalizability of OPS beyond NoMaD-style models.
The authors' rebuttal most of the concerns.

Given the positive consensus from the reviewers, I recommend acceptance.